# EMERGING PIXEL GROUNDING IN LARGE MULTIMODAL MODELS *Without* GROUNDING SUPERVISION

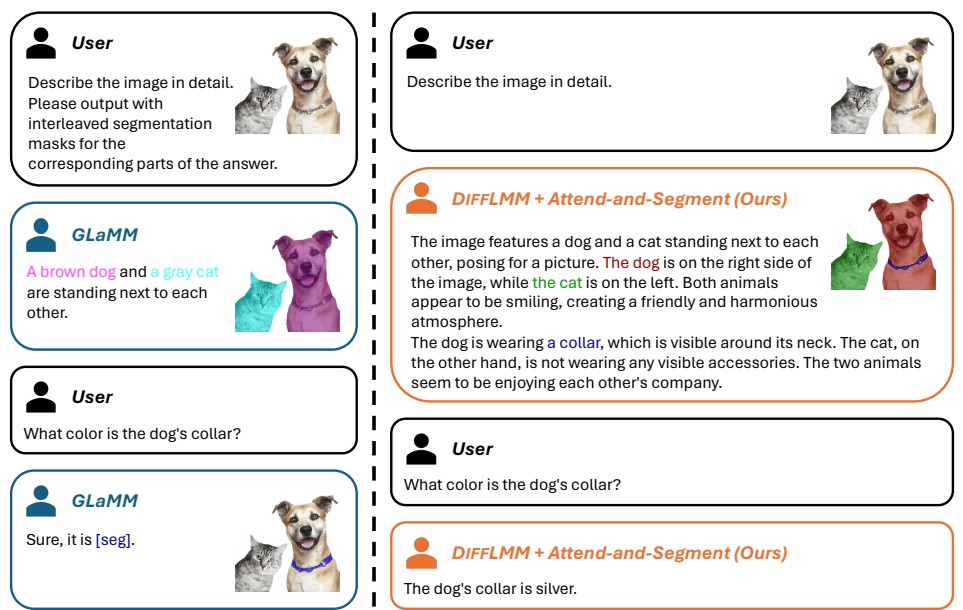

Figure 1: **Grounded conversations with GLaMM (**Rasheed et al., 2024**)** *vs*. **our approach,** DIFFLMM **+** *attend-and-segment***. Left:** As a state-of-the-art grounding LMM, GLaMM is trained to relate text phrases with segmentation masks while generating a response. However, due to limitations induced by the grounding supervision, it often fails to precisely follow the human user's instructions (*e.g.*, describing the image *in detail*, answering the correct *color*). **Right:** Our approach reveals and enhances the *grounding ability implicitly learned by LMMs without explicit grounding supervision*, which leads to visually grounded responses while preserving the general vision-language conversation ability of LMMs. More examples are shown in Figure 4.

## ABSTRACT

Current large multimodal models (LMMs) face challenges in grounding, which requires the model to relate language components to visual entities. Contrary to the common practice that fine-tunes LMMs with additional grounding supervision, we find that the grounding ability can in fact emerge in LMMs trained *without* explicit grounding supervision. To reveal this emerging grounding, we introduce an "*attend-and-segment*" method which analyzes the attention within standard LMMs to provide a point prompt to a segmentation model (*e.g.*, SAM) and perform pixel-level segmentation. Furthermore, to enhance the grounding ability, we propose DIFFLMM, an LMM utilizing a diffusion-based visual encoder, as opposed to the standard CLIP visual encoder, and trained with the same weak supervision. Without being constrained by the biases and limited scale of grounding-specific supervision data, our approach enables strong visual grounding while preserving general conversation abilities. We achieve competitive performance on both grounding-specific and general visual question answering benchmarks, compared with grounding LMMs and generalist LMMs, respectively. Notably, we achieve a 44.2 grounding mask recall on grounded conversation generation, outperforming the extensively supervised model GLaMM.

# 1 INTRODUCTION

Large multimodal models (LMMs) (Liu et al., 2023; Zhu et al., 2024; Dai et al., 2023) have brought the new opportunity of solving vision-language tasks in a general-purpose manner, which are typically built by connecting a visual encoder and a large language model (LLM) and fine-tuned by visual instructions. Currently, one major challenge faced by LMMs is *grounding*—the key ability of relating language components (*e.g.*, noun phrases) to visual entities (*e.g.*, objects) in a given image (Yu et al., 2016; Krishna et al., 2017). With the grounding ability, LMMs can lift the constraint of text-only responses and address more vision-language tasks in the real world.

To equip LMMs with the grounding ability, the common belief is that *additional supervision for grounding* is necessary, and corresponding architectural modifications need to be introduced. For instance, recent efforts extend the output modality from pure text to bounding boxes (Chen et al., 2023b; Peng et al., 2024), trace points (Xu et al., 2024), or segmentation masks (Lai et al., 2024; Rasheed et al., 2024), by 1) attaching additional modules to the vanilla LMM architecture, and 2) fine-tuning the LMM with grounding supervision. The grounding supervision originates from either re-purposing existing datasets that contain human-labeled object-level annotations or automatically annotating images using other models.

However, such *reliance on strong supervision* brings more undesired constraints: 1) *Limited training data*: Image datasets with high-quality object-level annotations (at most millions of images (Shao et al., 2019; Kuznetsova et al., 2020)) rely on pre-defined categories and are significantly smaller than those with coarse image-text pairs (up to billions (Schuhmann et al., 2022)), so re-purposing such object-level annotations only results in visual instruction data with limited diversity and quantity. Meanwhile, if the object-level annotations are produced by automated models, such annotations are noisier and less reliable than human-labeled ones (Rasheed et al., 2024). 2) *Supervision bias*: Changing the data focus to grounding tasks can lead to catastrophic forgetting (French, 1999) and hurt LMMs' general conversation capabilities. Furthermore, whether the grounding data are manually annotated (Lin et al., 2014) or pseudo-labeled by other models (Rasheed et al., 2024), they are biased by the annotators' or models' knowledge and may fail to align with general human preferences, as these fine-grained annotations can vary significantly among different annotators or models. 3) *Generalizability*: The grounding supervision is constrained within the visual concepts from either the existing datasets or other models, which contradicts with the ultimate goal of developing a general-purpose assistant for solving open-world problems (Bendale & Boult, 2015). Consequently, the resulting LMMs may *be biased by the limited grounding supervision data, generalize poorly to novel visual concepts and domains, and lose general conversation abilities.* Figures 1 and 4 show illustrative examples of these limitations.

To avoid such limitations, the question worth rethinking then arises: *Is there an approach to grounding LMMs other than strong supervision?* In fact, in this work, we reveal a critical yet previously overlooked fact: LMMs have inherently obtained the grounding ability through the weakly supervised visual instruction tuning. In other words, *the grounding ability can emerge in LMMs without grounding supervision.* Echoing prior observations of traditional convolutional neural networks (Zhou et al., 2015; 2016), we find that LMMs learn to detect visual entities and relate them with the language *implicitly*, during the progress of vision-language learning at the image level.

We therefore propose a simple and effective "*attend-and-segment*" strategy to *transform this emerging grounding ability into pixel-level segmentation masks*. Intuitively, the attention mechanism (Vaswani et al., 2017) in LMMs reveals *where the LMM is looking at*, and thus provides clues for visual grounding. We start with a base LMM trained without grounding supervision (Liu et al., 2023), and acquire its *attention corresponding to the visual input*. Though the entire attention map may be noisy, we locate the point where the LMM is focused on during token generation, and use the point to accurately prompt a segmentation model (Kirillov et al., 2023) for pixel-level grounding. With this *attend-and-segment* method, we enable vision-language tasks that directly rely on the grounding capability (*e.g.*, grounded conversation generation (Rasheed et al., 2024)). Remarkably, *attend-and-segment* does not require explicit grounding supervision like prior work; in contrast, *weak supervision* from standard visual instruction tuning data is sufficient to achieve performance comparable with or even higher than previous grounding-supervised models. Furthermore, as a general approach, *attend-and-segment* can be readily integrated with recent generalist LMMs (Li et al., 2024a; Tong et al., 2024a), and benefit from their stronger vision-language capabilities.

Furthermore, we introduce a simple solution to *enhance the emerging grounding ability of LMMs*. Previously, CLIP (Radford et al., 2021) plays a dominant role as the visual encoder of LMMs, due to its vision-language feature alignment. However, CLIP is known to be weak in providing localized visual features (Zhou et al., 2022; Ghiasi et al., 2022; Li et al., 2022), as its pre-training simply aligns the global representations of image-text pairs. Through systematic evaluation on both grounding-specific and general tasks, we find diffusion models (Ho et al., 2020; Rombach et al., 2022) a better alternative to CLIP, as their text-to-image generation capability enables *both vision-language alignment and localized features*. Thus, we propose the diffusion-based LMM (DIFFLMM), which augments the CLIP visual encoder of the LMM with a diffusion-based visual encoder, while being fine-tuned using the same data as the original LMM. To the best of our knowledge, DIFFLMM is the *first* successful integration of diffusion-based visual encoding and LMMs for both visual grounding and general vision-language tasks. Compared with the original LMM, DIFFLMM enhances the grounding ability without sacrificing performance in general-purpose vision-language tasks.

Our extensive experiments demonstrate that LMMs' grounding capabilities can *emerge from weak supervision*. Our approach, requiring no additional grounding supervision, *does not suffer from biases in the grounding supervision data, and generalizes better*. Despite being trained on less data than prior grounding LMMs (Lai et al., 2024; Rasheed et al., 2024), DIFFLMM achieves better or comparable performance on grounding-specific benchmarks, while adhering to a strong generalist model for vision-language tasks. To summarize, our contributions are three-fold:

- Different from prior methods that rely on grounding-specific strong supervision, we show the possibility of grounding LMMs without grounding supervision. Eliminating the need for fine-grained annotations from humans or external models, our approach is more scalable and generalizable.
- We discover a simple and effective approach, *attend-and-segment*, to achieve pixel-level grounding for LMMs by inspecting the attention mechanism in the generation process and prompting a segmentation model, which requires no grounding supervision or architectural changes.
- We propose DIFFLMM, which employs a visual encoder based on the diffusion model. DIFFLMM offers stronger grounding capabilities than the original LMM, while maintaining general vision-language task performance.

## 2 RELATED WORK

**Large multimodal models (LMMs).** Pioneering work in LMMs, such as LLaVA (Liu et al., 2023; Sun et al., 2024; Liu et al., 2024a;b), MiniGPT-4 (Zhu et al., 2024; Chen et al., 2023a), and InstructBLIP (Dai et al., 2023; Li et al., 2023a), enables visual inputs for large language models (LLMs) via vision-language feature alignment (Radford et al., 2021) and instruction tuning (Wei et al., 2022). To equip LMMs with the grounding ability, a series of methods have been proposed to produce model outputs of bounding boxes (Peng et al., 2024; Chen et al., 2023b; Wang et al., 2023; Pi et al., 2023; You et al., 2024; Li et al., 2024b), traces of points (Xu et al., 2024), or segmentation masks (Lai et al., 2024; Rasheed et al., 2024; Zhang et al., 2024; Ren et al., 2024), by adding region-specific tokens or decoders. These methods require further grounding supervision, so image datasets with fine-grained annotations (Lin et al., 2014; Yu et al., 2016; Zhou et al., 2017) are usually repurposed for the visual instruction tuning. Unlike these supervised methods, our approach, *attend-and-segment*, does not change the LMM architecture or require any grounding supervision data.

A concurrent work F-LMM (Wu et al., 2024a) shows a similar method to exploit attention maps in frozen LMMs for visual grounding, but we differ mainly in two aspects: 1) F-LMM follows the supervised paradigm to learn the segmentator, while our *attend-and-segment* requires *zero supervision*. For the first time, we reveal LMMs' emerging grounding capabilities without explicit supervision. 2) F-LMM examines existing LMMs without changing their visual encoding. In contrast, by analyzing visual encoders' grounding ability, we propose DIFFLMM to further enhance implicit grounding.

**Diffusion models (DMs) as visual feature extractors.** DMs (Song & Ermon, 2019; Ho et al., 2020; Song et al., 2021; Karras et al., 2022; Nichol & Dhariwal, 2021; Rombach et al., 2022) have become a prevalent paradigm in visual generation, and intermediate features from DMs are explored for applications beyond generative tasks. For example, DDPM-Seg (Baranchuk et al., 2022), ODISE (Xu et al., 2023), and EmerDiff (Namekata et al., 2024) utilize DM features for various segmentation tasks. Features from DMs can also establish point- or pixel-level correspondences between images (Tang

et al., 2023; Luo et al., 2023; Zhang et al., 2023b; Hedlin et al., 2023). For the first time, we show DMs can be utilized for learning a general-purpose LMM with strong grounding capabilities.

## 3 Approach

In this section, we first introduce the common architecture design of LMMs (Section 3.1). Then, we discuss *attend-and-segment*, which transforms the implicitly learned grounding ability into segmentation masks (Section 3.2). Based on the standard LMM and *attend-and-segment*, we propose DIFFLMM, to further enhance the grounding ability without additional supervision (Section 3.3). We include implementation details of our approach in Appendix A.

### 3.1 Preliminary: Meta-Architecture of Large Multimodal Models (LMMs)

Most LMMs (Liu et al., 2023; Zhu et al., 2024; Dai et al., 2023) share a common meta-architecture which consists of a visual encoder $M_V$, a vision-to-language feature projector $M_{V \mapsto L}$, and a large language model (LLM) $M_L$, as illustrated in Figure 2. Given an image $I$ of resolution $H \times W$, the visual encoder $M_V$ (*e.g.*, CLIP (Radford et al., 2021)) is employed to extract visual features $V = M_V(I) \in \mathbb{R}^{h \times w \times c_V}$, where $h \times w$ represents the feature map size, and $c_V$ is the visual feature dimension. Then, the visual feature map is considered as a sequence of $hw$ elements, and projected element-wise into the language feature space by the projector $M_{V \mapsto L}$. The projector can be implemented as a learnable lightweight multilayer perceptron (MLP). The $k$-th projected visual token is computed as $v_k = M_{V \mapsto L}(V_k) \in \mathbb{R}^{c_L}$, where $c_L$ is the feature dimension in the LLM. The visual tokens, concatenated with other language tokens, form the input sequence $S_{\text{input}}$:

$$S_{\text{input}} = \{t_1, \dots, t_p, v_1, \dots, v_{hw}, t_{p+1}, \dots, t_{p+q}\}, \tag{1}$$

where $\{v_1, \dots, v_{hw}\}$ are the $hw$ visual tokens projected from the visual feature map, $t_1, \dots, t_p$ are the $p$ language tokens before the visual tokens, and $\{t_{p+1}, \dots, t_{p+q}\}$ are the $q$ language tokens after the visual tokens.

The LLM is usually a decoder-only Transformer model, which is capable of next-token prediction. Given the input sequence $S_{\text{input}}$, the output sequence $S_{\text{output}} = \{o_1, \dots, o_r\}$ is generated in an auto-regressive manner, where the $i$-th token is predicted as:

$$o_i = M_L(S_{\text{input}}, o_1, \dots, o_{i-1}). \tag{2}$$

The generation is terminated when the last predicted token $o_r$ is a special "end-of-sequence" token.

### 3.2 *Attend-and-Segment*: Grounding LMMs Without Grounding Supervision

Prior efforts towards grounding LMM attach a detection or segmentation module to the LMM architecture, and specialize the LMM training procedure with grounding supervision, *i.e.*, visual instruction data augmented by object-level annotations, such that the LMM learns to predict connections between the text response and the image contents in the form of localized bounding boxes or segmentation masks. In contrast to these strongly supervised methods, we propose *attend-and-segment*, a simple and effective method for grounding LMMs *without changing their architecture or providing additional grounding supervision*. We investigate the attention maps inside the transformer-based language model when generating tokens, and observe strong interpretablity associated with the attention maps. Intuitively, the attention maps can provide information about *where the model is looking at* when producing outputs.

Formally, we consider the input token sequence $S_{\text{input}}$ as detailed in Section 3.1. When predicting an output token $o_i$, we capture the raw attention maps $A_i^{\text{raw}} \in [0, 1]^{n_{\text{layer}} \times n_{\text{head}} \times (p+hw+q+i-1)}$ inside the transformer-based LLM $M_L$, where $n_{\text{layer}}$ is the number of layers in the LLM, $n_{\text{head}}$ is the number of heads per layer, and $p + hw + q + i - 1$ is the number of tokens before the $i$-th output token $o_i$. We only use the attention maps associated with the $hw$ visual tokens, and reduce the dimensions by averaging over $n_{\text{layer}}$ layers and $n_{\text{head}}$ heads per layer. This operation returns an attention matrix $A_i^{\text{reduced}} \in [0, 1]^{h \times w}$, with the same spatial dimension as the visual feature map.

The attention between the output token and the visual tokens can provide interpretable grounding signals already. To further amplify the grounding signals and reduce the noise, we apply normalization

Figure 2: **Meta-architecture of LMMs and the *attend-and-segment* strategy.** In the standard LMM, the image encoder $M_V$ extracts visual features from an input image, and the features are transformed into visual tokens by the projector $M_{V \mapsto L}$. The large language model $M_L$ generates the output in an auto-regressive manner. When generating a new token (*e.g.*, "cat") which requires grounding, we capture the *attention* between the new token and the input visual tokens. Then a segmentation model (*e.g.*, SAM (Kirillov et al., 2023)) is prompted by the point with the highest attention value to produce a *segmentation mask* (*e.g.*, cat in the image).

across the whole output sequence:

$$A_i^{\text{norm}} = A_i^{\text{reduced}} - \frac{1}{r} \sum_{j=1}^{r} A_j^{\text{reduced}}, \tag{3}$$

where $r$ is the output sequence length.

To provide pixel-level grounding, we derive a segmentation mask by upsampling the attention map and prompting a pre-trained segmentation model (SAM (Kirillov et al., 2023) by default). For each token that requires grounding, we produce its corresponding binary mask by finding the point with the highest normalized attention and using its coordinate as a point prompt to the segmentation model. Thus, for elements of the output sequence, our *attend-and-segment* method provides pixel-level grounding results. Notably, we use off-the-shelf segmentation models without modification, while fine-tuning is inevitable in prior pixel-level grounding LMMs (Lai et al., 2024; Rasheed et al., 2024).

In downstream tasks like panoptic narrative segmentation, the tokens to ground are already provided, but other tasks like grounded conversation generation require the model to generate grounded noun phrases. In such tasks, we utilize existing natural language processing tools (*e.g.*, spaCy (Honnibal et al., 2020)) to parse the output sequence into noun phrases, and associate noun phrases with the output tokens. For each noun phrase, we produce segmentation masks using the average of normalized attention maps from the corresponding tokens.

### 3.3 DIFFLMM: ENHANCED GROUNDING WITH DIFFUSION-BASED LMM

Most LMMs employ CLIP (Radford et al., 2021) as the visual encoder because it has been pre-trained to align vision and language representations, but CLIP is known to be sub-optimal in tasks that require precise localization (*e.g.*, object detection, image segmentation) (Zhou et al., 2022; Ghiasi et al., 2022; Li et al., 2022). To enhance the grounding ability of LMMs, a direct choice may be replacing CLIP with better localized pure-vision backbones like DINO (Caron et al., 2021; Oquab et al., 2024). However, the lack of alignment with language representations can hurt vision-language task performance (Jiang et al., 2023; Tong et al., 2024b).

Compared with vision-language models with image-level alignment (*e.g.*, CLIP) and pure-vision models (*e.g.*, DINO), visual representations from diffusion models (DMs) strike a better balance: 1) DMs learn to generate high-fidelity images, for which well-localized visual features are necessary. Consequently, they are better than CLIP at localization. 2) DMs are trained to perform text-to-image generation, and in this procedure, they acquire alignment with language instructions, which is lacking in pure-vision models like DINO. Therefore, we propose diffusion-based LMM (DIFFLMM, illustrated in Figure 3), which strengthens the visual encoder with a pre-trained DM.

To extract visual features for a given input image $I$, we simulate one denoising step in the diffusion process. The image is tokenized by a vector quantized (VQ) encoder, added with a random noise,

Figure 3: **Visual encoding in DIFFLMM.** We perform one denoising step with the diffusion model (DM) (Ho et al., 2020; Rombach et al., 2022), and extract visual features from an intermediate block of the U-Net. The implicit captioner (Xu et al., 2023) produces text-like conditioning and improves the visual features in the U-Net. We combine both DM features and CLIP features, and add learnable positional encodings to them. The final visual features are projected into the language feature space, and fed into the LLM along with other text tokens. The DM and CLIP visual encoder are frozen.

and fed into the U-Net model of a DM (Ho et al., 2020; Rombach et al., 2022). We extract the visual feature map from the second upsampling block in the U-Net, which best preserves visual semantics (Tang et al., 2023). Text conditioning can enhance the visual feature extraction in the DM, but the image caption is usually unavailable. We employ the implicit captioning mechanism (Xu et al., 2023), which simulates text conditioning by the CLIP visual encoder. Specifically, the CLIP visual features are extracted as $V_{\text{CLIP}} = M_{\text{CLIP}}(I)$, projected by a multilayer perceptron (MLP) $M_{\text{CLIP} \mapsto \text{SD}}$, and injected into the U-Net via cross-attention modules. We denote the DM visual features as $V_{\text{SD}} = M_{\text{SD}}(I, M_{\text{CLIP} \mapsto \text{SD}}(V_{\text{CLIP}}))$. Finally, the visual feature map $V$ is composed by concatenating both DM features and CLIP features (note that we can reuse the CLIP features without additional overhead), and adding a set of learnable positional encodings $PE$ (Vaswani et al., 2017) to further enhance localization awareness:

$$V = \text{concat}(V_{\text{SD}}, V_{\text{CLIP}}) + PE \in \mathbb{R}^{h \times w \times c_V}. \tag{4}$$

For efficient training and preventing overfitting, we freeze pre-trained parameters in the CLIP visual encoder and the DM. Only the MLP in the implicit captioner, the positional encodings, and the vision-language feature projector are learnable in the visual encoder of DIFFLMM. Since the computation is dominated by the large language model component in DIFFLMM, integrating diffusion models in DIFFLMM does not significantly impact the efficiency. We only observe a marginal increase in the training and inference time ($< 5\%$).

## 4 EXPERIMENTS

In this section, we first present an analytical experiment to demonstrate the implicit grounding ability of LMMs (Section 4.1), and then show results of applying our approach in both visual grounding (Sections 4.2 and 4.3) and general conversation tasks (Section 4.4). Finally, we include ablation study of our module designs (Section 4.5). Due to limited space, we include implementation details and further results in the appendix. It is worth noting that *attend-and-segment* and DIFFLMM are general approaches for LMMs, but considering the computation limitations, we focus on grounding and enhancing LMMs with 7B or 8B-scale language models (Chiang et al., 2023; Meta, 2024).

### 4.1 PILOT STUDY: INSTANCE SEGMENTATION

We start by conducting an analytical study via *instance segmentation* (He et al., 2017) on MS-COCO (Lin et al., 2014) to demonstrate the emergence of grounding ability in LMMs and how different visual encoders impact this ability. Different from vision-language entangled benchmarks (which will be tested in later sections), the *vision-centric* instance segmentation task 1) directly focuses on relating image regions (represented as segmentation masks) with visual concepts (object categories), which is exactly the objective of grounding, and 2) does not evaluate based on language generation, making it more convenient to directly compare grounding abilities in different models.

Table 1: **Analysis of grounding abilities based on instance segmentation.** We examine the grounding ability embedded in the attention maps of LMMs, and compare LMMs trained with different visual backbones (including CLIP (Radford et al., 2021; Cherti et al., 2023), DINOv2 (Oquab et al., 2024), and Stable Diffusion (Rombach et al., 2022)) and the same data without grounding supervision, based on LLaVA-1.5 (Liu et al., 2024a). The original LLaVA-1.5 achieves a non-trivial performance compared with the baseline of randomly sampling points and prompting SAM. DIFFLMM enhances this grounding ability with diffusion-based visual features, and even surpasses Cambrian-1 (Tong et al., 2024a), which relies on an ensemble of four visual encoders, on mask AR.

| Model | Visual Backbone | **PAcc** | $AP_S$ | $AP_M$ | $AP_L$ | **AP** | $AR_S$ | $AR_M$ | $AR_L$ | **AR** |
|---|---|---|---|---|---|---|---|---|---|---|
| Random Point | | 10.53 | 0.0 | 0.2 | 0.8 | 0.3 | 0.1 | 1.2 | 10.1 | 3.8 |
| LLaVA-1.5 | CLIP (original) | 34.01 | 1.8 | 6.6 | 6.3 | 3.9 | 5.8 | 21.7 | 43.2 | 22.8 |
| | ConvNeXt CLIP | 37.16 | **3.1** | 7.0 | 8.4 | 4.9 | **8.4** | 22.1 | 44.0 | 23.9 |
| | DINOv2 | 34.55 | 1.9 | 6.7 | 7.2 | 4.2 | 6.4 | 22.0 | 41.7 | 23.0 |
| **DIFFLMM** | SD-1.5 | 38.92 | 2.1 | 7.6 | **9.9** | **5.7** | 6.4 | 25.3 | **48.8** | 25.9 |
| **(Ours)** | SD-1.5 + CLIP | **40.22** | 1.6 | **7.9** | 9.6 | 5.6 | 6.3 | **25.5** | 47.3 | **26.0** |
| Cambrian-1 | Ensemble | 44.49 | 2.0 | 6.9 | 10.6 | 6.0 | 6.3 | 20.7 | 39.1 | 21.4 |

LMMs are not originally designed for instance segmentation. Therefore, for evaluation purposes, we ask LMMs to generate a detailed description of a given image, and leverage *attend-and-segment* to produce pairs of noun phrases and segmentation masks from the LMM response. Then we find the best-matching category label for each noun phrase by computing their embedding similarities using spaCy (Honnibal et al., 2020). Since the LMM is not constrained to only describe objects that are annotated by the dataset (and should not be rewarded or penalized for detecting out-of-domain objects), we exclude predictions that cannot be matched with any category label that appear in the given image. We compare the standard metrics in instance segmentation, mask average precision (AP) and mask average recall (AR). In this setting, AP is lower than AR because the models are not supervised for the task, and we do not explicitly remove duplicated predictions. To assess the quality of point prompts derived from LMM attention maps without SAM, we compute a new metric, point accuracy (PAcc), which is the ratio of point prompts that correctly fall into the masks of the corresponding category. For comparison, we consider a baseline that simulates a "blind" LMM, which prompts SAM with a random point for segmenting each ground-truth category.

As shown in Table 1, the attention-derived prompts from the original LLaVA-1.5 achieve a non-trivial 34.01 accuracy, indicating that the attention maps can be utilized for fine-grained grounding. Further performing segmentation with SAM leads to 22.8 AR. Comparing models equipped with different visual encoders but trained with the same data, our DIFFLMM achieves the best overall point accuracy and mask AP/AR, whether or not we concatenate the diffusion features with CLIP features. A recent vision-centric LMM, Cambrian-1 (Tong et al., 2024a), utilizing an ensemble of four visual encoders, has an even higher point accuracy and mask AP. However, due to different training data, it tends to generate shorter descriptions than LLaVA-1.5, resulting in more missed objects and lower mask AR.

## 4.2 GROUNDED CONVERSATION GENERATION

The pilot study on instance segmentation shows that LMMs trained without explicit grounding supervision already implicitly acquires grounding ability, which can be used to produce pixel-level segmentation masks. Following the discussion above, we examine LMMs' grounding ability on a more comprehensive benchmark, grounded conversation generation (GCG) (Rasheed et al., 2024). The objective of GCG is to understand visual entities in an image, and organize them into a localized description. To be specific, the GCG task requires the LMM to generate a detailed caption for a given image, in which phrases are related to their corresponding segmentation masks in the image.

Since the GCG task requires model abilities in both captioning and segmentation, three types of metrics are considered: 1) To measure the caption quality, the *text-only metric*, METEOR (Banerjee & Lavie, 2005), compares the generated captions with the human-annotated reference captions. 2) To assess the segmentation mask quality, the *mask-only metric*, mean intersection-over-union (mIoU), quantifies the similarity between ground-truth masks and their matched predicted masks. 3) The

Table 2: **Grounded conversation generation (GCG) results.** Even without grounding supervision, *attend-and-segment* (*a&s* in the table) unlocks the implicitly learned grounding ability in LLaVA-1.5 (Liu et al., 2024a), *outperforming all grounding-specific models on this task*. DiffLMM further enhances the grounding ability, and leads to stronger grounding performance. The higher METEOR scores demonstrate our better preserved conversation ability. As a general approach, *attend-and-segment* can be applied on different LMMs (Li et al., 2024a; Tong et al., 2024a). All methods are evaluated by the text-only metric METEOR (M) (Banerjee & Lavie, 2005), the mask-only metric mIoU, and the combined metric grounding mask recall (Rec) (Rasheed et al., 2024) on the Grand$_f$ dataset (Rasheed et al., 2024). Baseline results are from GLaMM (Rasheed et al., 2024).

| Model | Grounding Supervision | Validation Set | | | Test Set | | |
|---|---|---|---|---|---|---|---|
| | | M↑ | mIoU↑ | **Rec↑** | M↑ | mIoU↑ | **Rec↑** |
| BuboGPT (Zhao et al., 2023) | | 17.2 | 54.0 | 29.4 | 17.1 | 54.1 | 27.0 |
| Kosmos-2 (Peng et al., 2024) | ✓ | 16.1 | 55.6 | 28.3 | 15.8 | 56.8 | 29.0 |
| LISA (Lai et al., 2024) | | 13.0 | 62.0 | 36.3 | 12.9 | 61.7 | 35.5 |
| GLaMM (Rasheed et al., 2024) | | 16.2 | **66.3** | 41.8 | 15.8 | **65.6** | 40.8 |
| LLaVA-1.5 + *a&s* + SAM (Ours) | | **18.6** | 58.0 | 44.2 | **18.3** | 59.3 | 42.7 |
| LLaVA-NeXT + *a&s* + SAM (Ours) | ✗ | 15.6 | 64.5 | 45.6 | 15.6 | **65.6** | **44.2** |
| Cambrian-1 + *a&s* + SAM (Ours) | | 14.6 | 59.8 | 42.0 | 14.5 | 60.7 | 40.4 |
| **DiffLMM + *a&s* + SAM (Ours)** | | 18.4 | 61.2 | **46.6** | 18.2 | 62.1 | **44.2** |

grounding mask recall (Rasheed et al., 2024) is an *integrated metric* for region-specific grounding, which considers both the mask IoU and the textual similarities between the predictions and the ground truth. Therefore, the grounding mask recall is mainly considered when comparing different models.

In Table 2 we compare our approach, which learns the LMM without any grounding supervision, with prior methods for grounding LMMs (Zhao et al., 2023; Peng et al., 2024; Lai et al., 2024; Rasheed et al., 2024). *Even without grounding supervision*, our *attend-and-segment* leads to 42.7 mask recall for the original LLaVA-1.5 (Liu et al., 2024a), which is already *higher than all the previous grounding LMMs*. As a general approach, *attend-and-segment* can be used in conjunction with recent LMMs such as LLaVA-NeXT (Li et al., 2024a) and Cambrian-1 (Tong et al., 2024a), and benefit from their improved visual encoding and vision-language capabilities. Compared with CLIP-based LMMs, DiffLMM provides better localized visual features and improves the grounding ability. When using our DiffLMM as the LMM, we reach the highest 44.2 test recall. Our method achieves pixel grounding but does not suffer from the supervision bias brought by grounding annotations, and thus better preserves the text-only conversation abilities, as shown by the higher METEOR scores.

## 4.3 Referring Expression Segmentation and Panoptic Narrative Grounding

In addition to GCG, we provide additional results on two widely investigated visual grounding tasks: **referring expression segmentation (RES)** (Hu et al., 2016; Yu et al., 2016), which requires segmenting a target object specified by a given referring expression, and **panoptic narrative grounding (PNG)** (González et al., 2021), which grounds each noun phrase in a given text description with a panoptic segmentation mask. We employ two other segmentation models, Co-DETR (Zong et al., 2023) and OpenSeeD (Zhang et al., 2023a), in RES and PNG, respectively. The details of our implementation and complete results are included in Appendix B.

We would like to note that **directly comparing our approach with prior supervised grounding LMMs is not exactly fair**, due to the following reasons: 1) Our approach requires *no grounding supervision*, while all prior grounding LMMs are extensively trained on such grounding tasks. 2) In both task settings, *the text for grounding is set by an external input*, which is inconsistent with the generative nature of LMMs. Grounding LMMs have to be trained to generate a special token for mask decoding, but our approach does not include such training. Therefore, a conversation between a human user and an LMM has be simulated for indirectly producing the attention maps and segmentation results. Considering both factors, RES and PNG are inherently challenging for our approach without grounding supervision. Nevertheless, we achieve competitive performance on these two tasks, setting a new state of the art for zero-shot approaches, as shown in Tables 3 and 4.

Table 3: **Results on referring expression segmentation (RES).** We use Co-DETR (Zong et al., 2023) for class-agnostic mask proposal generation. Without RES supervision, *attend-and-segment* enhances the ability to localize corresponding objects for given referring phrases, compared with prior zero-shot methods. All models are evaluated by the average cumulative intersection-over-union (cIoU) on RefCOCO(+/g) (Yu et al., 2016). 'Sup?' indicates whether grounding supervision on RES is used.

| Model | Sup? | cIoU |
|---|---|---|
| LISA (Lai et al., 2024) | | 69.9 |
| GROUNDHOG (Zhang et al., 2024) | ✓ | 74.2 |
| GLaMM (Rasheed et al., 2024) | | **75.6** |
| LLaVA-1.5 + F-LMM (Wu et al., 2024a) | | 69.0 |
| Global-Local CLIP (Yu et al., 2023) | | 26.5 |
| TAS (Suo et al., 2023) | | 32.5 |
| Ref-Diff (Ni et al., 2023) | ✗ | 36.1 |
| LLaVA-1.5 + *a&s* + Co-DETR (Ours) | | 35.2 |
| DIFFLMM + *a&s* + Co-DETR (Ours) | | **37.0** |

Table 4: **Results on panoptic narrative grounding (PNG).** We use a specialized panoptic segmentation model, OpenSeeD (Zhang et al., 2023a) to generate candidate masks. Our approach achieves competitive results even without grounding supervision. DIFFLMM improves the original LLaVA-1.5 for visual grounding, which is consistent with our results on other tasks (Tables 1 and 2). The metric is average recall. 'Sup?' indicates whether grounding supervision on PNG is used.

| Model | Sup? | AR |
|---|---|---|
| PixelLM[†] (Ren et al., 2024) | | 43.1 |
| GLaMM[†] (Rasheed et al., 2024) | ✓ | 55.8 |
| GROUNDHOG (Zhang et al., 2024) | | **66.8** |
| LLaVA-1.5 + F-LMM (Wu et al., 2024a) | | 64.8 |
| DatasetDiffusion[‡] (Nguyen et al., 2023) | | 23.5 |
| DiffSeg[‡] (Tian et al., 2024) | | 24.1 |
| DiffPNG (Yang et al., 2024) | ✗ | 38.5 |
| LLaVA-1.5 + *a&s* + OpenSeeD (Ours) | | 42.2 |
| DIFFLMM + *a&s* + OpenSeeD (Ours) | | **45.3** |

[†]: Reported by F-LMM. [‡]: Reported by DiffPNG.

## 4.4 VISUAL QUESTION ANSWERING

When enhancing the grounding ability of LMMs, we do not want LMMs to lose their general vision-language abilities. To assess such general abilities, we evaluate DIFFLMM on a wide range of visual question answering (VQA) benchmarks, including VQAv2 (Goyal et al., 2017), GQA (Hudson & Manning, 2019), Vizwiz (Gurari et al., 2018), ScienceQA-IMG (Lu et al., 2022), and TextVQA (Singh et al., 2019). We also consider more comprehensive LMM benchmarks, including POPE (Li et al., 2023b), MMBench (Liu et al., 2024c), and LLaVA-Bench (Liu et al., 2023).

It is worth noting that previous grounding LMMs (*e.g.*, LISA (Lai et al., 2024), GLaMM (Rasheed et al., 2024)) are not usually evaluated on these general-purpose VQA benchmarks. For example, POPE is designed for quantifying object hallucination in LMMs by asking questions like "Is there an [object] in the image?" but the queried object often does not exist. However, we find that GLaMM almost always answers "Sure, it is [seg]." and provides an incorrect segmentation mask (see examples in Figure 4). Such loss of capabilities in answering general questions is due to *supervision bias*—these LMMs are fine-tuned for grounding tasks and they forget how to answer general visual questions without grounding. Therefore, grounding LMMs like GLaMM have extremely low scores on these benchmarks, and we choose to compare with stronger generalist LMMs that are not designed for grounding tasks on VQA benchmarks.

When compared with state-of-the-art LMMs of the same scale (fine-tuned from a 7B LLM), including InstructBLIP (Dai et al., 2023), IDEFICS (HuggingFace, 2023), Qwen-VL-Chat (Bai et al., 2023), and LLaVA-1.5 (Liu et al., 2024a), DIFFLMM ranks 1st on 3 benchmarks, and 2nd on 4 benchmarks. Since DIFFLMM is trained on the same data as LLaVA-1.5, similar results are observed. Therefore, our diffusion-based DIFFLMM improves fine-grained vision-language comprehension that specifically requires the grounding ability, while retaining strong general vision-language capabilities.

## 4.5 ABLATION STUDY

Our *attend-and-segment* applies normalization across the sequence of attention maps (Equation 3), which significantly reduces noises in the maps (Figure 7). From the attention map, we select the single point with the highest attention value to prompt SAM, instead of providing the entire map as a mask prompt. Empirically, we find that attention maps are sparse, tending to focus on a few key points within objects rather than the entire objects, so point prompts are more effective. Quantitative comparisons are summarized in Table 6. We use point-based prompts in all other experiments.

Table 5: **Visual Question Answering (VQA) results.** We evaluate and compare generalist LMMs of the same scale (all with a 7B-sized LLM) on a wide range of benchmarks, including VQAv2 (Goyal et al., 2017), GQA (Hudson & Manning, 2019), Vizwiz (VW) (Gurari et al., 2018), ScienceQA-IMG (SQA) (Lu et al., 2022), TextVQA (TQA) (Singh et al., 2019), POPE (Li et al., 2023b), MMBench (MM-B) (Liu et al., 2024c), and LLaVA-Bench (LV-B) (Liu et al., 2023). Different from prior models, DIFFLMM is built upon a diffusion model (DM) visual encoder, which provides stronger grounding (Tables 1 and 2) and preserves vision-language abilities in general tasks. Notably, GLaMM (Rasheed et al., 2024) fails in these general VQA tasks. For each benchmark, the **1st** and 2nd best models are marked. Baseline results are from LLaVA-1.5 (Liu et al., 2024a).

| Model | Visual | VQAv2 | GQA | VW | SQA | TQA | POPE | MM-B | LV-B |
|---|---|---|---|---|---|---|---|---|---|
| InstructBLIP (Dai et al., 2023) | CLIP | - | 49.2 | 34.5 | 60.5 | 50.1 | 78.9 | 36.0 | 60.9 |
| IDEFICS (HuggingFace, 2023) | CLIP | 50.9 | 38.4 | 35.5 | - | 25.9 | - | 48.2 | - |
| Qwen-VL-Chat (Bai et al., 2023) | CLIP | 78.2 | 57.5 | 38.9 | 68.2 | **61.5** | - | 60.6 | - |
| LLaVA-1.5 (Liu et al., 2024a) | CLIP | **78.5** | 62.0 | **50.0** | 66.8 | 58.2 | **85.9** | 64.3 | **65.4** |
| **DIFFLMM (Ours)** | DM | 78.3 | **62.1** | 48.1 | **69.3** | 57.2 | 85.7 | **66.2** | 63.7 |

Table 6: **Ablation study on *attend-and-segment*.** Normalizing attention maps across the entire sequence removes noisy patterns and improves grounding. Prompting SAM (Kirillov et al., 2023) with a single point instead of a low-resolution mask is more effective. Our *attend-and-segment* combines both techniques. The results are based on evaluating DIFFLMM on the GCG task (Rasheed et al., 2024).

| Attn Norm | SAM Prompt | GCG Val mIoU↑ | Rec↑ |
|---|---|---|---|
| ✓ | Mask | 50.0 | 36.5 |
| ✗ | Point | 57.4 | 44.1 |
| ✓ | Point | **61.2** | **46.6** |

Table 7: **Ablation study on general conversations of DIFFLMM.** We pre-train DIFFLMM on the data as LLaVA-1.5 (Liu et al., 2024a), and compare the converged losses of various backbones including CLIP (Radford et al., 2021), DINOv2 (Oquab et al., 2024), and SD-1.5 (Rombach et al., 2022). Lower losses indicate better vision-language alignment. Both positional encodings (PE) and implicit captioner (IC) improve the convergence of DIFFLMM, and support it to preserve general conversation abilities.

| Model | Backbone | PE | IC | Loss↓ |
|---|---|---|---|---|
| LLaVA-1.5 | CLIP | – | | **2.027** |
| | DINOv2 | – | | 2.403 |
| | CLIP+DINOv2 | – | | 2.088 |
| **DIFFLMM (Ours)** | SD-1.5 | | | 2.384 |
| | SD-1.5 | ✓ | | 2.338 |
| | SD-1.5 | ✓ | ✓ | 2.141 |
| | SD-1.5+CLIP | ✓ | ✓ | **2.032** |

In DIFFLMM (Figure 3), we employ a few modules to enhance the visual feature extraction and *alignment with the language model.* Specifically, we 1) add learnable *positional encodings* (Vaswani et al., 2017) to the visual features, and 2) use the *implicit captioner* (Xu et al., 2023) to simulate text conditioning with CLIP visual features. Due to limited computation, we cannot retrain models with the full dataset of LLaVA-1.5 (Liu et al., 2024a) and run thorough evaluation as in the previous sections. Instead, we examine the modules' effects with respect to the optimization objective in the pre-training stage (Liu et al., 2023), as summarized in Table 7. Trivially replacing the CLIP (Radford et al., 2021) visual encoder with DINOv2 (Oquab et al., 2024) leads to significantly higher losses, which implies worse vision-language alignment. Thanks to the text-to-image training, SD-1.5 (Rombach et al., 2022) results in a smaller loss. The positional encodings close the loss gap by about 13%, and further adding the implicit captioner reduces the gap by another 55%.

## 5 CONCLUSION

In this work, we reveal a previously overlooked yet critical fact that LMMs possess grounding capabilities even if they are trained *without* grounding supervision. We propose *attend-and-segment* to convert this implicit grounding ability into segmentation masks, and we introduce DIFFLMM to further enhance the grounding ability. Our approach is more scalable and generalizable compared with supervised methods. Moreover, extensive evaluation results demonstrate strong performance on both grounding-specific and general vision-language benchmarks, even surpassing grounding LMMs trained with extensive supervision on the challenging grounded conversation generation task.

**Ethics statement.** Despite the promising results, our approach shares several common limitations with other LMMs, particularly regarding the visual instruction tuning data and pre-training models. Social biases may be introduced during data collection, filtering, and annotation processes. Therefore, our approach should be used with cautious in human-related applications, and we aim to develop LMMs with improved alignment and safety in the future.

**Reproducibility statement.** We ensure reproducibility by providing detailed implementation descriptions and data specifications in Appendix A. Additionally, our code is included in the supplementary material. We will publicly release the code upon the acceptance of this work.

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

## A  IMPLEMENTATION DETAILS OF OUR APPROACH

In this section, we provide the implementation details of this work to ensure reproducibility of our experiments.

***Attend-and-Segment.*** We first collect the attention maps for the visual tokens, and aggregate the attention maps by averaging over all layers and heads. Then we apply normalization across the output token sequence to remove noisy points, and upsample the normalized attention map to the original image resolution. During mask refinement, we find the coordinate where the normalized attention value is maximized, and use it as a prompt to the ViT-H SAM model (Kirillov et al., 2023) for producing the pixel-level segmentation map. In the instance segmentation and grounded conversation generation tasks, we parse the model response into noun phrases using `spaCy` (Honnibal et al., 2020), and for each phrase, we average the normalized attention maps for the tokens that compose the central noun of the phrase.

**DIFFLMM.** Our development of DIFFLMM is based on the codebase and dataset of LLaVA-1.5 (Liu et al., 2024a). We employ the Stable Diffusion v1.5 (Rombach et al., 2022) model as our visual backbone. In the denoising step, we add a random noise at the 100 timestep, and extract features from the second upsampling block, following the practice of DIFT (Tang et al., 2023). We also provide additional ablation study on the choice of the noise level and feature block in Table 8. In the implicit captioner (Xu et al., 2023), we employ the visual encoder of CLIP-ViT-L-336px (Radford et al., 2021), which is the same CLIP model in the original LLaVA-1.5. The model is trained with LoRA (Hu et al., 2022), and the same training recipe as LLaVA-1.5. The training data are also the same as LLaVA-1.5. The included datasets and their licenses are listed below.

- LAION (Schuhmann et al., 2022): MIT License.
- CC (Changpinyo et al., 2021): "The dataset may be freely used for any purpose, although acknowledgement of Google LLC ("Google") as the data source would be appreciated."
- SBU (Ordonez et al., 2011): Unknown license.
- MS-COCO (Lin et al., 2014): Creative Commons Attribution 4.0 International License.
- GQA (Hudson & Manning, 2019): Creative Commons Attribution 4.0 International License.
- OCR-VQA (Mishra et al., 2019): Unknown license.
- TextVQA (Singh et al., 2019): Creative Commons Attribution 4.0 International License.
- VisualGenome (Krishna et al., 2017): Creative Commons Attribution 4.0 International License.

Table 8: **Ablation study on diffusion feature extraction.** Adding a relatively small noise (at diffusion step 100 or 200) to the original image and extracting features from the second upsampling block in the diffusion U-Net lead to the best results in DIFFLMM.

| Noise Step | Feature Block | Pre-train Loss ↓ |
|:---:|:---:|:---:|
| 100 | 2 | **2.384** |
| 0 | 2 | 2.417 |
| 200 | | 2.395 |
| 300 | | 2.457 |
| 100 | 1 | 2.400 |
| | 3 | 2.465 |
| | 4 | 2.625 |

## B  DETAILS OF REFERRING EXPRESSION SEGMENTATION AND PANOPTIC NARRATIVE GROUNDING

**Referring expression segmentation (RES).** The RES task requires the model to segment a target object specified by a given referring expression. It is indirect for our method, *attend-and-segment*,

to provide a single attention map for the referring expression specified by a task input. We design a simple strategy to find the text-image correspondence. Specifically, we formulate a one-round conversation between a human user and an LMM. For a referring expression `[expr]`, the human user asks the model to "`Describe the [expr].`" Then, the model is expected to generate a response that focuses on the target object. We find the first token of the object of the sentence (*e.g.*, the token right after the verb "is"), and extract its attention map. After that, we can apply *attend-and-segment* to find the point with the highest attention value, and produce a grounding mask based on it. An illustrative example is shown as follows, in which we will use the token "**a**" for attention map extraction.

```
USER: Describe the "middle player."
MODEL: The "middle player" in the image is a baseball player
wearing a blue shirt and a baseball glove...
```

We observe that SAM (Kirillov et al., 2023) tends to produce overly fine-grained segmentation masks for object parts, and cannot deliver satisfactory results in the RES task which requires masks for entire objects. Since our approach is actually *not tied with any specific segmentation model*, we can use the prompt point generated by *attend-and-segment* to guide other models for segmentation. In this experiment, we use a Co-DETR (Zong et al., 2023) instance segmentation model to produce class-agnostic mask predictions for each image. To avoid data contamination, we exclude RefCOCO(+/g) validation/test images from the training set, and retrain the Co-DETR model. After generating both the prompt point with *attend-and-segment* and the set of candidate masks, we select the mask that contains the prompt point. If there are multiple masks containing the prompt point, we just use the mask with the highest confidence score predicted by Co-DETR.

The results are summarized in Table 9. Without any training on RES or other visual grounding tasks, we achieve a remarkable performance of 37.0 average cIoU in RES. In particular, we outperform the previous zero-shot methods (Yu et al., 2023; Suo et al., 2023; Ni et al., 2023). The results demonstrate strong visual grounding capabilities implicitly learned by LMMs.

Table 9: **Comprehensive results on the referring expression segmentation (RES) task.** Our approach uses Co-DETR (Zong et al., 2023) for class-agnostic mask proposal generation. Although DIFFLMM is trained without referring segmentation data, it significantly enhances the ability to localize corresponding objects for given referring phrases. *attend-and-segment* leads to improved segmentation quality than prior zero-shot methods (Yu et al., 2023; Suo et al., 2023; Ni et al., 2023), and DIFFLMM further enhances this grounding ability. All models are evaluated by the cumulative intersection-over-union (cIoU) metric.

| Model | Grounding Supervision | RefCOCO | | | RefCOCO+ | | | RefCOCOg | | Avg. |
|---|---|---|---|---|---|---|---|---|---|---|
| | | val | testA | testB | val | testA | testB | val | test | |
| LISA (Lai et al., 2024) | | 74.9 | 79.1 | 72.3 | 65.1 | 70.8 | 58.1 | 67.9 | 70.6 | 69.9 |
| GROUNDHOG (Zhang et al., 2024) | ✓ | 78.5 | 79.9 | 75.7 | 70.5 | 75.0 | **64.9** | 74.1 | 74.6 | 74.2 |
| GLaMM (Rasheed et al., 2024) | | **79.5** | **83.2** | **76.9** | **72.6** | **78.7** | 64.6 | **74.2** | **74.9** | **75.6** |
| LLaVA-1.5 + F-LMM (Wu et al., 2024a) | | 75.2 | 79.1 | 71.9 | 63.7 | 71.8 | 54.7 | 67.1 | 68.1 | 69.0 |
| Cropping (Yu et al., 2023) | | 22.7 | 21.1 | 23.1 | 24.1 | 22.4 | 23.9 | 28.7 | 27.5 | 24.2 |
| Global-Local CLIP (Yu et al., 2023) | | 24.9 | 23.6 | 24.7 | 26.2 | 24.9 | 25.8 | 31.1 | 31.0 | 26.5 |
| TAS (Suo et al., 2023) | | 29.5 | 30.3 | 28.2 | 33.2 | 38.8 | 28.0 | 35.8 | 36.2 | 32.5 |
| SAM-CLIP (Ni et al., 2023) | ✗ | 25.2 | 25.9 | 24.8 | 25.6 | 27.8 | 26.1 | 33.8 | 34.8 | 28.0 |
| Ref-Diff (Ni et al., 2023) | | 35.2 | 37.4 | 34.5 | 35.6 | 38.7 | **31.4** | **38.6** | **37.5** | 36.1 |
| LLaVA-1.5 + *a&s* (Ours) | | 38.8 | 47.0 | 34.2 | 32.9 | 40.0 | 27.3 | 31.8 | 29.6 | 35.2 |
| DIFFLMM + *a&s* (Ours) | | **41.9** | **48.0** | **37.1** | **34.1** | **40.1** | 30.0 | 32.9 | 31.8 | **37.0** |

**Panoptic narrative grounding (PNG).** The PNG task requires the model to ground each noun phrase in a given text description of a given image, by providing a corresponding panoptic segmentation mask (Kirillov et al., 2019) for each noun phrase. Different from the standard setup of LMMs, the text description of the image is provided by the task, rather than generated by the model itself. Therefore, we adapt our *attend-and-segment* method for the PNG task. Specifically, we simulate a one-round conversation between a human user and an LMM. The human user asks the model to "`Describe the image in detail.`" Then, the model responds with the given text description. We extract attention maps from the model response part of this conversation, and use the attention maps to guide the segmentation procedure.

Similar to RES, we use a specialized panoptic segmentation model, OpenSeeD (Zhang et al., 2023a), to provide high-quality panoptic segmentation masks for each image. For each noun phrase, we find the first associated token and its attention maps. Then, we locate the point with the highest attention value, and select the mask that contains this point, from all the candidate masks generated by OpenSeeD. Note that in panoptic segmentation, all masks are non-overlapping, so there is only one mask that contains this point of the highest attention value. This selected mask is predicted as the region corresponding to the noun phrase.

In Table 10, we compare the results of our approach with prior grounding LMMs. Although our approach is applied on the PNG task in a zero-shot manner, it achieves competitive performance and even outperforms one prior grounding LMM, PixelLM. Additionally, our approach improves over all previous zero-shot PNG methods (Nguyen et al., 2023; Tian et al., 2024; Yang et al., 2024).

Table 10: **Comprehensive results on the panoptic narrative grounding (PNG) task.** Our approach uses a specialized panoptic segmentation model, OpenSeeD (Zhang et al., 2023a) to generate candidate masks, and applies *attend-and-segment* on the LMM-generated attention maps. Our approach achieves competitive results even without grounding supervision. DIFFLMM improves the original LLaVA-1.5 for visual grounding, which is consistent with our results on other tasks. The metric is average recall. The results of PixelLM and GLaMM are reported by F-LMM (Wu et al., 2024a). The results of DatasetDiffusion and DiffSeg are reported by DiffPNG (Yang et al., 2024).

| Model | Grounding Supervision | All | Thing | Stuff |
|---|:---:|---|---|---|
| PixelLM (Ren et al., 2024) | | 43.1 | 41.0 | 47.9 |
| GLaMM (Rasheed et al., 2024) | | 55.8 | 52.9 | 62.3 |
| GROUNDHOG (Zhang et al., 2024) | ✓ | **66.8** | **65.0** | **69.4** |
| LLaVA-1.5 + F-LMM (Wu et al., 2024a) | | 64.8 | 63.4 | 68.2 |
| DatasetDiffusion (Nguyen et al., 2023) | | 23.5 | 16.0 | 33.8 |
| DiffSeg (Tian et al., 2024) | | 24.1 | 17.7 | 33.0 |
| DiffPNG (Yang et al., 2024) | ✗ | 38.5 | 36.0 | 42.0 |
| LLaVA-1.5 + *a&s* (Ours) | | 42.2 | 34.6 | 52.9 |
| DIFFLMM + *a&s* (Ours) | | **45.3** | **38.7** | **54.5** |

## C  QUALITATIVE RESULTS

In Figure 4 we present qualitative results of DIFFLMM + *attend-and-segment* for more challenging visual questions that are different from the training data, in comparison with GLaMM (Rasheed et al., 2024). First, when the questions are not formulated as usual, GLaMM tends to interpret these questions as image captioning or referring segmentation tasks, while DIFFLMM can still follow the user's instructions and accurately answer the questions. Meanwhile, *attend-and-segment* provides well-grounded responses that connects text phrases and visual entities. Furthermore, our approach shows *better generalizability to unfamiliar* question types, visual concepts, and image domains.

We present additional qualitative results for the grounded conversation generation task in Figure 5. The DIFFLMM model is asked to "Describe the image in detail." Then we use *attend-and-segment* to produce visual grounding. Overall, our approach can provide accurate segmentation masks, but may also suffer from common issues of LMMs (*e.g.*, object hallucination (Li et al., 2023b; Sun et al., 2024)).

## D  ADDITIONAL ANALYSIS OF ATTENTION MAPS

In *attend-and-segment*, we aggregate the attention values between each generated token and the visual tokens into a 2D map. In this section, we provide more in-depth analysis of the attention maps. For visualization, we use the same "cat and dog" image (Figure 1) as an example in the following analysis; we have similar observations on other images as well.

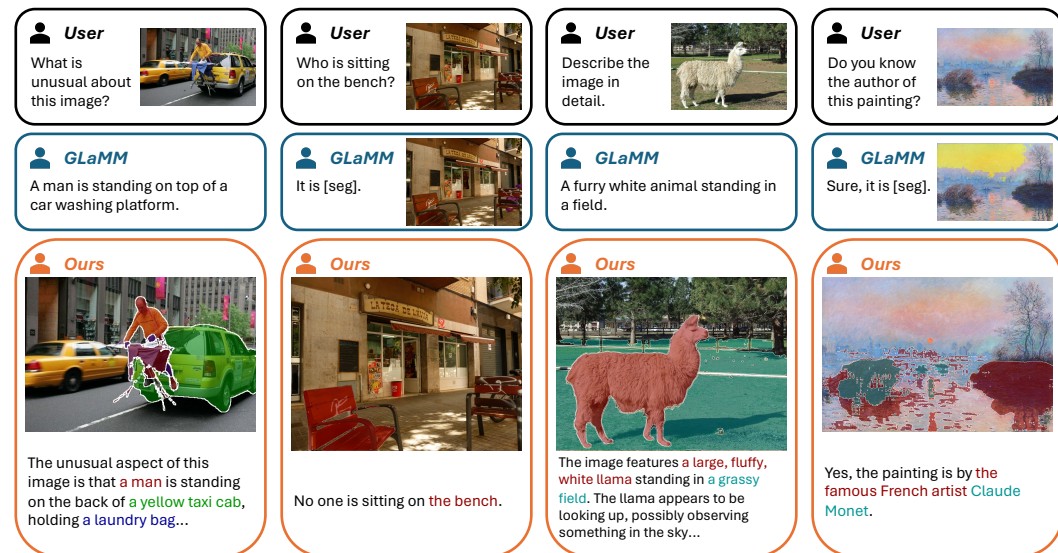

Figure 4: **Comparison of model responses to challenging visual questions.** 1) *Unusual image contents*: The model is requested to analyze the unusual aspect of a given image. Compared with GLaMM, our approach provides a more detailed and accurate answer with grounding. 2) *Adversarial questions*: The model is asked about something that does not exist in the image. GLaMM insists to segment the bike behind the bench in this example. 3) *Rare visual concepts*: The image contains objects of less frequent categories. In this example, GLaMM does not recognize the llama but describes it in a general manner, while our approach provides a more accurate description. 4) *Shifted image domain*: An image from a new domain is given to the model. Interestingly, our approach seems to be making the decision based on the texture and style in the painting. For visual clarity, we only show the beginning parts of our model responses if they are too long. These challenging examples demonstrates better *generalizability of our approach*.

**Attention in each head and layer.** Instead of averaging the attention values over $n_{\text{layer}}$ layers and $n_{\text{head}}$ heads per layer in the LLM, we first inspect the individual attention values in each head and layer. Figure 6 visualizes the attention between one generated token "cat" and the input visual tokens. Consistent with some recent observations (Wu et al., 2024b), a few heads in the intermediate layers show stronger activation with respect to the visual object in the image. Also, attention maps in intermediate layers are more localized. However, it is infeasible to build direct connections between attention heads and visual concepts, given the absence of grounding annotations.

Table 11 summarizes an empirical study that demonstrates the grounding results of using the attention from one single head of one single layer. Compared with averaging over all heads and layers, individual heads and layers lead to to significantly worse and noisier results. Therefore, we aggregate the attention maps across all heads and layers by averaging, which also simplifies the algorithm of *attend-and-segment* in our setting without grounding supervision.

**Attention normalization.** After reducing the attention maps into one 2D map for each generated token, we observe some noisy patterns in the attention maps (Figure 7-top). Some seemingly uninformative visual tokens (usually in the background) attracts more attention from the generated token than other visual tokens. A recent work (Darcet et al., 2024) shows similar observations, and explains that such less informative tokens are "repurposed for internal computations." To remove such artifacts, they propose to provide additional tokens to the Vision Transformer as registers.

However, in our setting, we cannot retrain the visual backbone or the language model due to limited data and computes. Instead, we simply normalize the attention maps by subtracting the mean attention map averaged over the output sequence (Section 3.2). Although the noisy attention patterns exist, we observe that these patterns are relatively stable (Figure 7-top), so the mean attention map, aggregated over the output sequence, can capture the undesired attention patterns and allow us to remove them.

Table 11: **Evaluation of attention maps from individual head/layer combinations.** Applying *attend-and-segment* on the attention maps extracted from individual heads and layers results in worse and less stable grounding mask recall in GCG, as compared with applying *attend-and-segment* on the mean attention maps aggregated over all heads and layers, which achieves 46.6 mask recall (Table 2).

| | | Head Index | | | | avg.±std. |
|---|---|---|---|---|---|---|
| | | 1 | 9 | 17 | 25 | |
| | 1 | 19.2 | 13.6 | 19.9 | 11.9 | 16.2±3.5 |
| Layer | 9 | 7.5 | 25.1 | 9.0 | 28.2 | 17.5±9.3 |
| Index | 17 | 26.2 | 4.3 | 19.7 | 27.1 | 19.3±9.1 |
| | 25 | 5.9 | 34.3 | 27.0 | 15.7 | 20.7±10.8 |
| Overall | | | | | | 18.4±8.8 |

After the attention normalization, we observe clearer patterns (Figure 7-bottom) which leads to accurate pixel grounding. Quantitatively, attention normalization improves the GCG mask recall from 44.1 to 46.6 (Table 6). In addition to noun phrases, other words reveal relations or comparisons between visual entities, and could be helpful for more vision-language tasks. We leave this investigation for future research.

**Visualization of attention maps.** We visualize the attention maps in DIFFLMM in complex scenes in Figure 8, to demonstrate the ability to differentiate similar visual entities and understand complex phrases. In Figure 9, we examine the attention maps inside the diffusion U-Net, which uses a CLIP text encoder. Due to the limited capacity of the text encoder, the attention maps produced by the diffusion U-Net are ineffective for visual grounding.

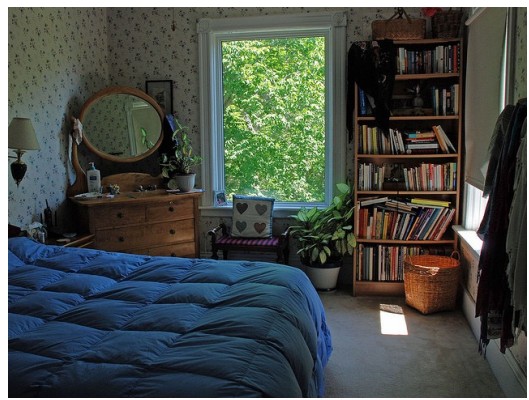 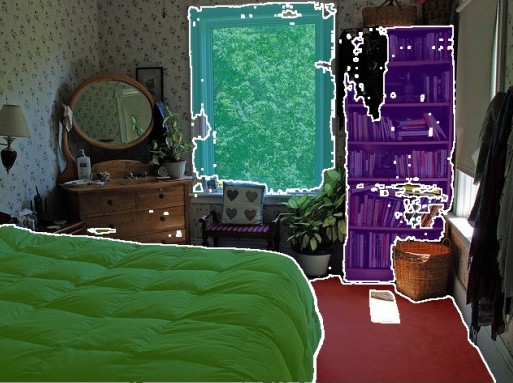

The image features a cozy bedroom with a large bed situated in the center of the room. The bed is covered with a blue comforter, and it is positioned next to a window that allows natural light to enter the room. The room also has a dresser and a mirror, adding to the overall comfort and functionality of the space.

In addition to the bed, there are several books scattered throughout the room, indicating that the occupant may be an avid reader. A potted plant is placed near the window, adding a touch of greenery to the room. A vase can also be seen on the dresser, further enhancing the room's decor.

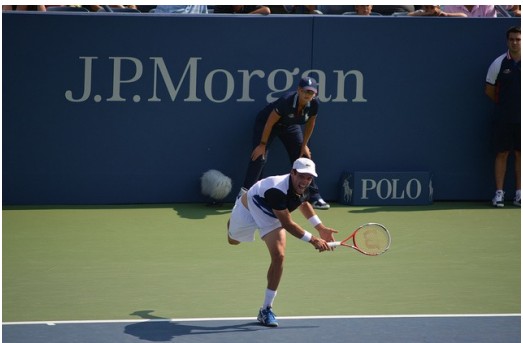 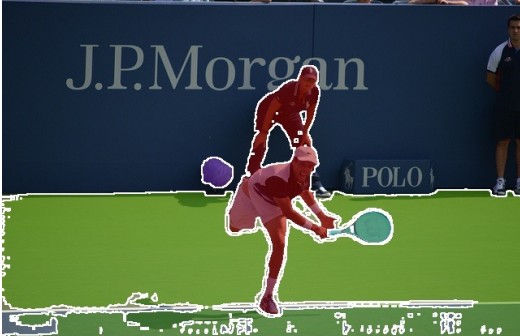

The image captures a tennis match in progress, with a man in a white shirt and white shorts playing on a blue tennis court. He is in the middle of a swing, holding a tennis racket and preparing to hit the ball. The tennis ball is visible in the air, close to the player. There are several other people in the scene, likely spectators or fellow players. Some of them are standing near the edges of the court, while others are positioned further away. A chair can be seen on the side of the court, possibly for resting or observing the match.

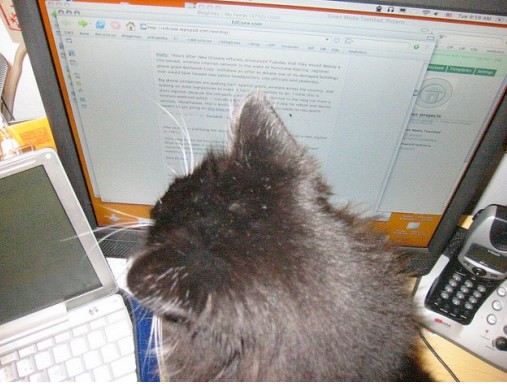 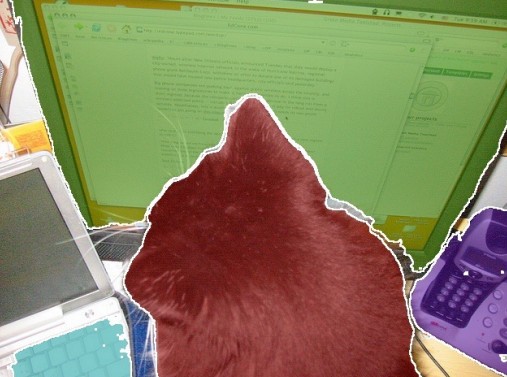

The image features a black cat sitting in front of a computer monitor, which is displaying a webpage. The cat appears to be looking at the screen, possibly intrigued by the content. The computer setup includes a keyboard placed to the left of the monitor and a cell phone on the right side.

In addition to the cat and the computer setup, there is a person visible in the background, likely the owner of the cat or someone working in the same space.

Figure 5: **Qualitative results for grounded conversation generation.** For visual clarity, we only display the best four non-overlapping segmentation masks per image.

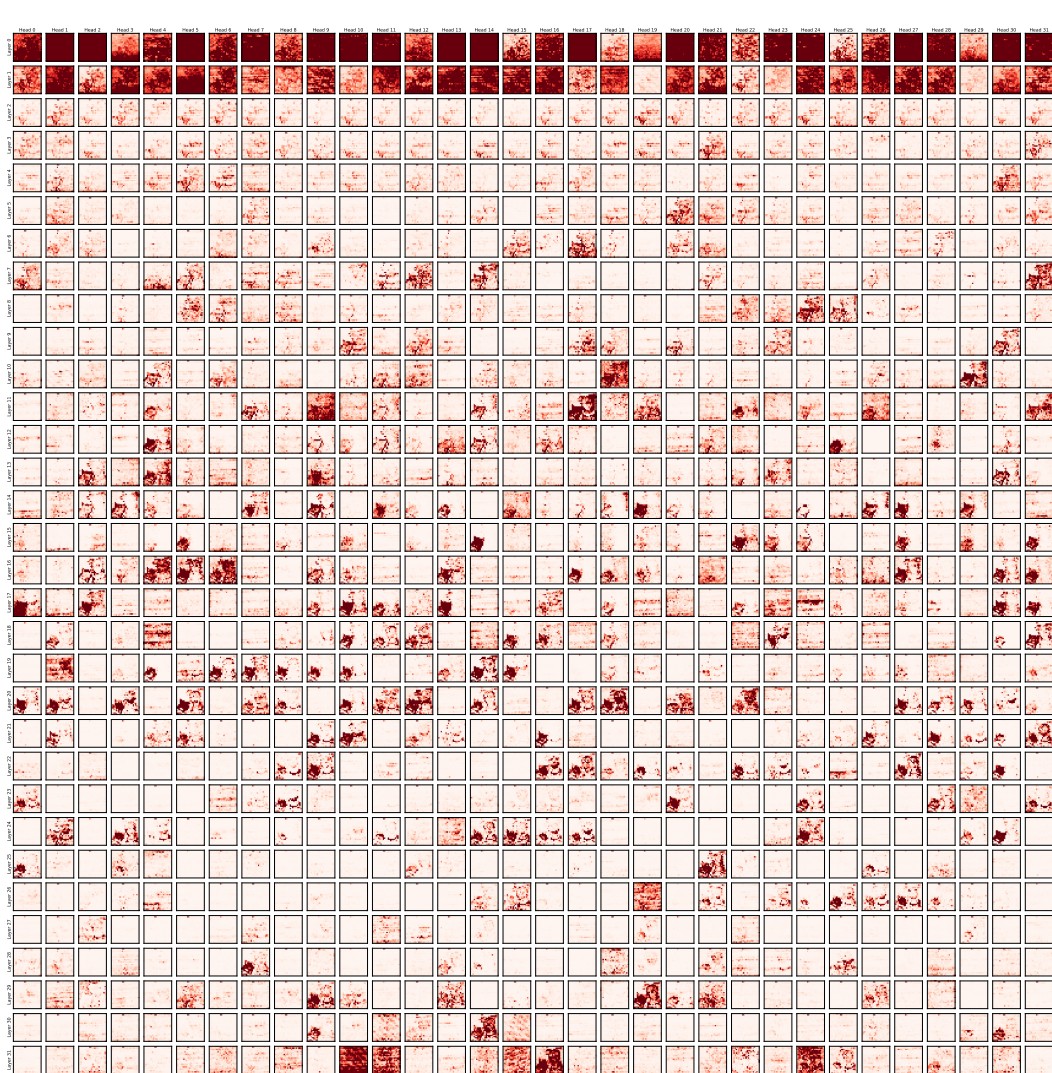

Figure 6: **Attention between the visual tokens and the generated token "cat."** We observe certain heads in the intermediate layers produce more localized attention maps with respect to the "cat" object in the image (*e.g.*, Head 14 of Layer 15). It remains challenging to directly relate individual heads to visual concepts when grounding annotations are not available, so *attend-and-segment* directly aggregates attention maps from all layers and heads by averaging them.

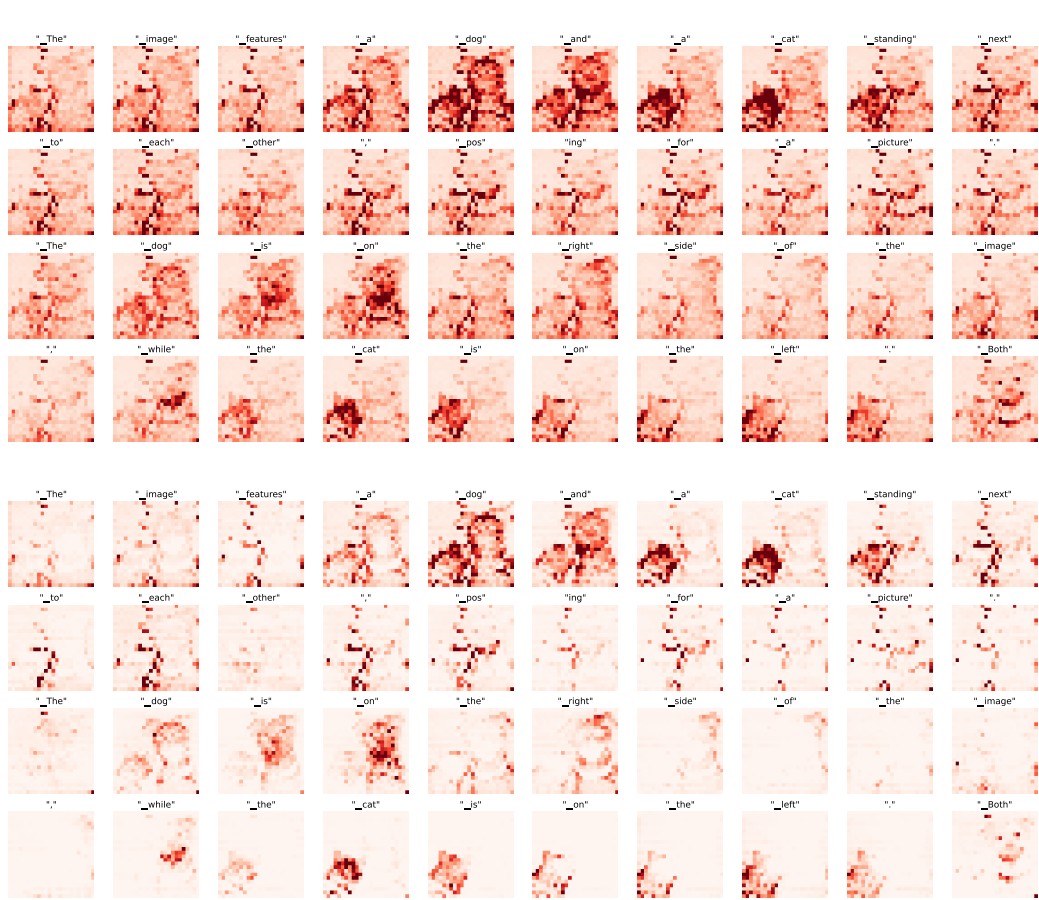

Figure 7: **Attention maps before and after the normalization.** Top: Before the normalization, a few uninformative visual tokens in the background (*e.g.*, top-center tokens above the dog's head) receive more attention, which is consistent with recent observations (Darcet et al., 2024). Such patterns are stable across the output sequence. Bottom: To remove such artifacts in the attention maps, we subtract the mean attention map (Section 3.2). After the normalization, the attention maps show clearer localization, and are suitable for pixel-level grounding. In addition to noun phrases, other parts of the text response demonstrate meaningful visual correspondence (*e.g.*, "next to each other" and the space between the two animals).

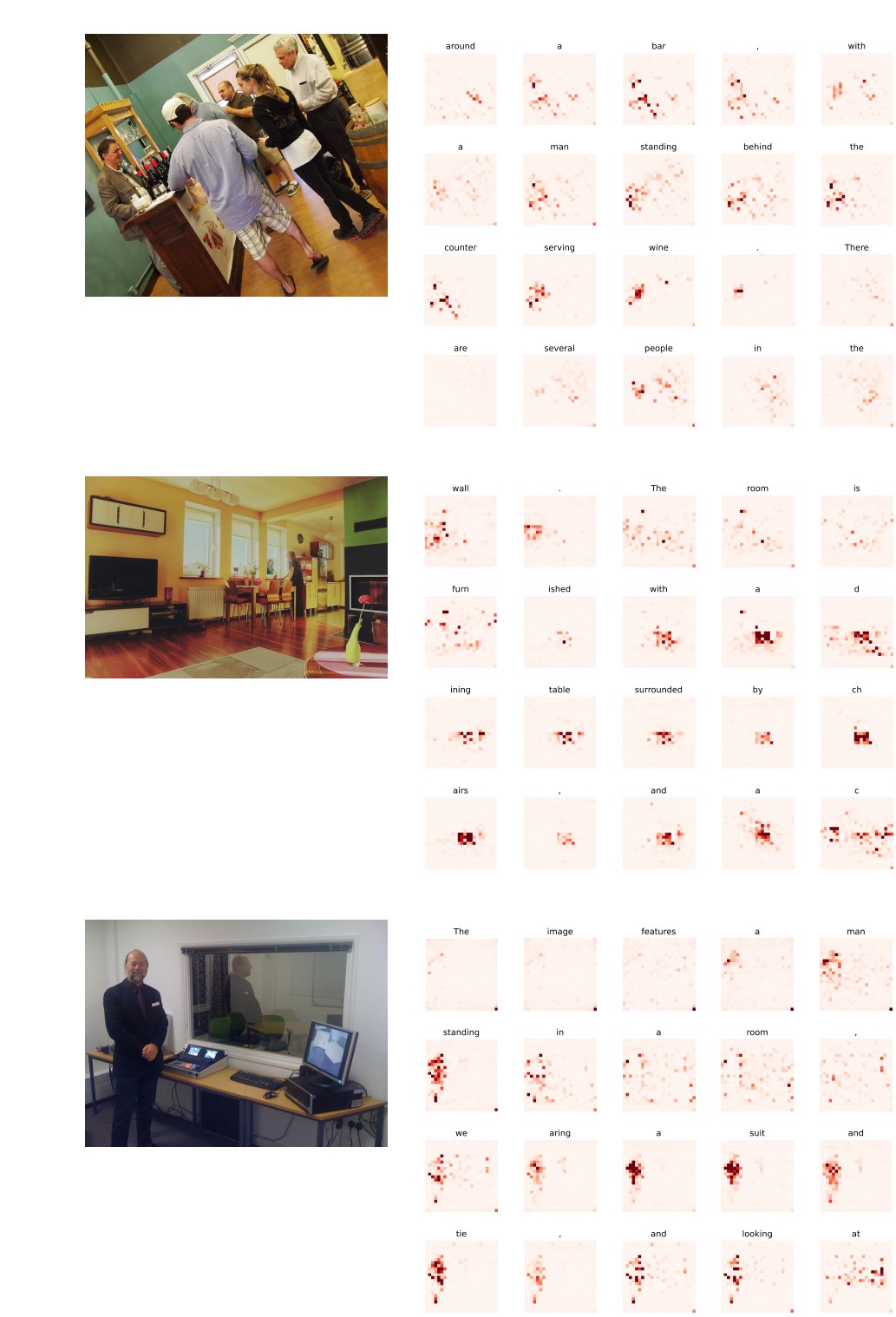

Figure 8: **Attention maps in complex scenes.** Our approach is able to reason and ground complex phrases (*e.g.*, "a man standing behind the counter serving wine") via the attention maps in the LMM. In complex scenes with other similar objects (*e.g.* other men in the first image, another table on the left side of the second image, and the man in the mirror in the third image), our approach can still correctly locate the target object by finding the point with the highest attention value.

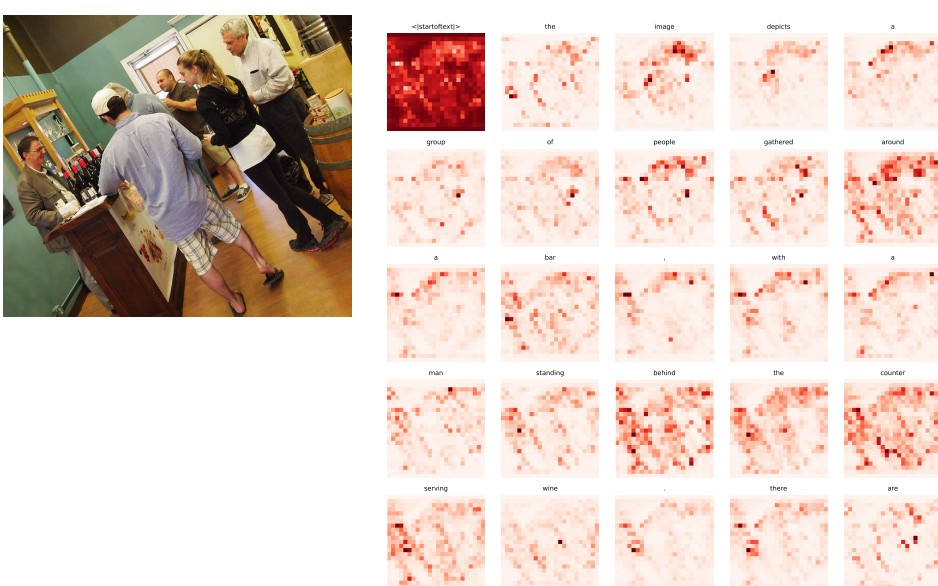

Figure 9: **Attention maps from the diffusion U-Net.** The U-Net in the diffusion model also computes cross-attention between image patches and a given text condition. However, the attention maps are more noisy compared with the attention maps in the LMM (see Figure 8), and thus are less effective in visual grounding.

