# OpenReview forum: "Emerging Pixel Grounding in Large Multimodal Models Without Grounding Supervision"
_ICLR.cc/2025/Conference — Submitted to ICLR 2025_

### Official Review · Reviewer_4QCn · 2024-10-30

**Soundness:** 2
**Presentation:** 3
**Contribution:** 2
**Rating:** 5
**Confidence:** 5

**Summary:**

The paper shows the possibility of grounding LMMs without grounding supervision. The paper presents a method called attend-and-segment that can be applied to existing LMMs to achieve pixel level grounding. The paper introduces DIFFLMM, which employs a visual encoder based on the diffusion model. The experiments are conducted on Grounded Conversation Generation (GCG) and VQA benchmarks.

**Strengths:**

1. The paper reads smooth and is mostly clear.

2. The motivation of using attention maps for pixel-level grounding is reasonable.

3. The proposed DiffLMM achieves good experimental results compared to CLIP-based, and DINO-based methods for GCG, while maintaining general VQA performance.

**Weaknesses:**

1. From experiments Table4 and Figure2, it seems that the quality of the attention maps are not good enough for grounding. Please also report the result without SAM. And it seems the model highly relies on SAM, a foundation CV model, to produce good segmentation mask. So the sentence "the grounding ability can **emerge** in LMMs without grounding supervision. " is a little bit over-claimed. In addition, using SAM may also conflict with the claim "without explicit grounding supervision", as SAM is trained with a huge number of grounding related data.

2. Lack of experiments on referring expression segmentation (RES) benchmark [a,b] and panoptic narrative grounding (PNG) benchmark [c]. RES benchmarks can better validate the effectiveness of the proposed method, as they require LMMs to perform both reasoning and segmentation. PNG benchmark validates the ability of LMMs to segment user-described objects and ground key phrases or words in a text sequence.

3. As pointed out in the paper, the paper is mostly related to  F-LMM (Wu et al., 2024a), however, no direct experimental comparisons are provided.

4. Experimental results in Table1 shows that the AP and AR of DIFF-LMM are very low (only 5.6 AP and 26.0 AR), limiting its application in real scenarios.

[a] Kazemzadeh S, Ordonez V, Matten M, et al. Referitgame: Referring to objects in photographs of natural scenes[C]//Proceedings of the 2014 conference on empirical methods in natural language processing (EMNLP). 2014: 787-798.
[b] Mao J, Huang J, Toshev A, et al. Generation and comprehension of unambiguous object descriptions[C]//Proceedings of the IEEE conference on computer vision and pattern recognition. 2016: 11-20.
[c] González C, Ayobi N, Hernández I, et al. Panoptic narrative grounding[C]//Proceedings of the IEEE/CVF International Conference on Computer Vision. 2021: 1364-1373.

**Questions:**

1. The method utilizes existing natural language processing tools (e.g., spaCy (Honnibal et al., 2020)) to parse the output sequence into noun phrases. However, in some cases only "noun phrases" may fail to capture the whole semantic meaning in some complex scenario, e.g. "the man in black holding an umbrella". How to handle this problem?

---

> ### Author Response · Authors · 2024-11-25
> **Response to Reviewer 4QCn (Part 1)**
>
> We appreciate the detailed feedback you provided for our submission. We are encouraged by your acknowledgement of our paper writing, our “motivation of using attention maps for pixel-level grounding is reasonable,” our “good experimental results” for GCG, and “maintaining general VQA performance.” We provide the following clarifications in response to your concerns:
>
> *W1. From experiments Table4 and Figure2, it seems that the quality of the attention maps are not good enough for grounding. Please also report the result without SAM. And it seems the model highly relies on SAM, a foundation CV model, to produce good segmentation mask. So the sentence "the grounding ability can emerge in LMMs without grounding supervision. " is a little bit over-claimed. In addition, using SAM may also conflict with the claim "without explicit grounding supervision", as SAM is trained with a huge number of grounding related data.*
>
> - In our attend-and-segment method, we only need **one point** with the highest attention value from the attention map to provide a clue for grounding, so we **do not require the entire attention map to be precise**. As long as the attention map is able to produce a reasonable starting point for SAM as prompt, our method can generate accurate pixel-level grounding. This simple design also minimizes the number of hyper-parameters. Otherwise, for example, we would need thresholding hyper-parameters to convert the real-valued attention maps into binary masks.
>
> - We already provided results **without SAM** in Table 1. The point accuracy (PAcc) metric measured how accurately the prompt points (derived from attention maps) could be used to locate objects of interest.
>
> - We would like to clarify that this “no grounding supervision” claim is proper.
>
>     - “Grounding” is the ability to relate language with visual contents, as previously defined in the literature [Ref1, Ref2]. We would like to note that SAM is trained with **semantic-free** mask annotations, i.e., SAM can perform the segmentation task, but has **no grounding ability**.
>
>     - In contrast, previous grounding LMMs are extensively trained with grounding supervision that includes correspondences between text and fine-grained visual contents (e.g., object bounding boxes and segmentation masks).
>
>     - Our approach only requires **image-level** weak supervision, so it is proper to claim that "the grounding ability can emerge in LMMs without grounding supervision.”
>
> We hope this response could clarify the reviewer’s concern, and we would be glad to discuss further details if the reviewer has more questions.
>
> *W2. Lack of experiments on referring expression segmentation (RES) benchmark [a,b] and panoptic narrative grounding (PNG) benchmark [c]. RES benchmarks can better validate the effectiveness of the proposed method, as they require LMMs to perform both reasoning and segmentation. PNG benchmark validates the ability of LMMs to segment user-described objects and ground key phrases or words in a text sequence.*
>
> We greatly appreciate your suggestions on additional evaluation on other tasks. We have performed experiments on referring expression segmentation (RES) and panoptic narrative grounding (PNG) and achieved stronger performance compared with prior zero-shot methods that are also not trained with explicit grounding supervision. For example, our approach achieves 45.3 average recall in PNG, outperforming the previous zero-shot state-of-the-art method by 6.8 average recall. The details are included in Section D of the updated manuscript. The results on RES and PNG further demonstrate that LMMs can implicitly learn to accurately perform visual grounding without explicit grounding supervision, and our approach is the key to unlock such grounding ability.

---

> ### Author Response · Authors · 2024-11-25
> **Response to Reviewer 4QCn (Part 2)**
>
> *W3. As pointed out in the paper, the paper is mostly related to F-LMM (Wu et al., 2024a), however, no direct experimental comparisons are provided.*
>
> - First of all, we would like to point out that we are not required to compare with a concurrent and unpublished arXiv paper, according to ICLR policy (https://iclr.cc/Conferences/2025/ReviewerGuide).
>
> - Since F-LMM does not provide results on the instance segmentation or grounded conversation generation (GCG) task, we were unable to provide a direct comparison between our approach and F-LMM. In the updated manuscript, we have included additional comparison with F-LMM on RES and PNG.
>
> - Nevertheless, we discussed the differences between our approach and F-LMM in the related work section, especially in the type of supervision and the visual encoding.
>
> *W4. Experimental results in Table1 shows that the AP and AR of DIFF-LMM are very low (only 5.6 AP and 26.0 AR), limiting its application in real scenarios.*
>
> - In fact, the real-world scenario was demonstrated in the GCG task (Table 2), in which we have **outperformed the previous grounding LMMs** like GLaMM.
>
> - We would like to clarify that the results in Table 1 were about analytical experiments on the instance segmentation task, which provided insights that attention maps in LMMs could lead to visual grounding. We did not claim it as a powerful instance segmentation approach in real-world application.
>
> *Q1. The method utilizes existing natural language processing tools (e.g., spaCy (Honnibal et al., 2020)) to parse the output sequence into noun phrases. However, in some cases only "noun phrases" may fail to capture the whole semantic meaning in some complex scenario, e.g. "the man in black holding an umbrella". How to handle this problem?*
>
> - The LMM is able to understand complex phrases like what the reviewer has suggested, because the large language model (LLM) component can correctly reason the correspondence between elements in complex phrases and visual contents.
>
> - This correct correspondence, as reflected in the attention maps, will not be changed by how we parse the sentences.
>
> - We have updated the manuscript with additional visualization of the attention maps for phrases in complex scenes (Figure 8), which demonstrates the correct correspondence between objects and parts of such complex phrases.
>
> [Ref1] Rohrbach et al. Grounding of Textual Phrases in Images by Reconstruction. ECCV 2016.
>
> [Ref2] Krishna et al. Visual genome: Connecting language and vision using crowdsourced dense image annotations. IJCV 2017.

---

> > ### Comment · Reviewer_4QCn · 2024-11-27
> > **Reviewer Response**
> >
> > Thanks for the detailed feedback from the authors, which resolves some of my concerns. It is encouraged to add these discussions in the revised paper. However I believe that the paper still requires a major revision before publication. There are still two major concerns remained:
> >
> > 1) The authors argue that "the grounding ability can [emerge] in LMMs without grounding supervision." However, from the experiments, we see that the  grounding ability of existing LMMs is too weak to produce good visual grounding results and still highly relies on SAM to improve visual grounding. So I still believe there are some sentences in the abstract and introduction sections over-claiming.
> >
> > 2) The method utilizes existing natural language processing tools (e.g. SpaCy) to parse noun phrases, which simply enumerates all noun words and phrases from a sentence. (1) From Fig.8, we can also find that the parsing process can be quite noisy. For example, in the case of "a man standing behind the counter serving wine", the attention maps of the word 'man' is not focusing on the target person, instead, all people in the room are activated. This concern is also related to the accuracy of the produced attention maps. (2) The usage of SpaCy also makes direct comparisons with prior works unfair, which can automatically find which target to focus on.

---

> > > ### Author Response · Authors · 2024-11-28
> > > **Response to Reviewer 4QCn Follow-up (Part 1)**
> > >
> > > Thank you for your continued and thoughtful feedback. We are encouraged that we have addressed most of the reviewer’s previous concerns. We appreciate your engagement with our work and the opportunity to address your remaining concerns in the paper presentation.
> > >
> > > - Regarding the reviewer’s concern about the emerging grounding ability, we would like to clarify:
> > >     - The implicit grounding ability of existing LMMs is already strong enough to induce **point-level grounding, even without SAM**. Visual grounding, which “relates language components to visual entities” (Line 40 in Abstract), can be implemented in various forms, including boxes, traces, and masks (e.g., as in Kosmos-2, PixelLLM, and LISA, respectively). In particular, a recent LMM Molmo [Ref3] focuses on point-level grounding and enables various applications such as object counting and robot manipulation. In our work, we first reveal the implicit grounding by relating tokens with points (via attention), and then use SAM or other segmentation models to extend the point prompt into pixel-level masks. As shown in the PAcc metric of Table 1, existing LMMs (e.g., LLaVA-1.5, Cambrian-1) already achieve the goal of point-level grounding.
> > >
> > >     - The use of SAM is semantic-free, which is **only for producing the segmentation and not for improving grounding correspondence**. Many existing visual grounding tasks require producing a segmentation mask, so we employ segmentation models in the tasks. However, segmentation models could not improve the direct correspondence between vision and language. If the point prompt is not correctly located on the target object, SAM could not fix it. Without the correct point-level grounding directly from LMMs’ attention, we could not achieve strong pixel-level grounding in visual grounding tasks (Tables 2-4).
> > >
> > >     - We have revised the abstract and introduction to accurately describe the process of point prompting via LMM attention values and the role of segmentation models (e.g., SAM) in pixel-level grounding. For example, we revised Line 100 in Section 1 as: though the entire attention map may be noisy, we only need the point with the highest attention value for accurate point-level grounding, and subsequently, use the point prompt for pixel-level grounding via a segmentation model.
> > >
> > > - Regarding the reviewer’s concern about the parsing process, we would like to clarify:
> > >     - In the visualization of Figure 8 (“Attention maps in complex scenes”), the attention map of the word “man” is indeed focusing on the correct target person:
> > >         - Note that attention values, computed via softmax, are always **non-negative**. Even if we apply normalization (Equation 3), it is inevitable that some attention is distributed to regions that are not exactly the target object. Furthermore, due to semantic relationships between the objects (e.g., all the people in the room), we do observe some attention on other people in the room, but such values are **significantly lower** (5x smaller) than the attention on the person on the left side of the image. The point of the highest attention value is correctly located on the target person, and this one point, instead of the entire attention map, is exactly what our attend-and-segment method uses for prompting segmentation.
> > >         - In contrast, Figure 9 (“Attention maps from the diffusion U-Net”) shows an example of attention maps that are not suitable for visual grounding. Even if we find the point with the highest attention value in Figure 9, it cannot locate the correct target object.
> > >         - Another viable strategy would be filtering the attention map with a pre-defined threshold, but it would introduce additional hyper-parameters. Our “max-attention point” strategy is simpler and more effective, as we have shown in Table 6 of the ablation study.
> > >         - We have revised the caption of Figure 8 to incorporate the results of our discussion on sentence parsing.

---

> ### Author Response · Authors · 2024-11-28
> **Response to Reviewer 4QCn Follow-up (Part 2)**
>
> - Regarding the reviewer’s concern about the parsing process, we would like to clarify (continued):
>     - Our method is still **fully automated**, consistent with prior grounding LMMs. Our comparison with prior works **is fair, except the supervision** (prior works require grounding supervision, while ours does not).
>         - In our work, we do not rely on any explicit grounding supervision to distinguish words that require grounding, but such words are required in the GCG task. Therefore, we take a minimalist perspective, and **adopt a simple NLP tool spaCy to parse noun phrases without human involvement**.
>         - The usage of spaCy is **not always required**. For example, the RES and PNG tasks already provide the noun phrase to ground, in which cases we do not need spaCy to parse sentences.
>         - Even in GCG, in principle, our method is not tied with spaCy, and a more advanced and automatic approach could have been used. For instance, in our response to Reviewer r6Gw, we showed an example of using a large language model, GPT-4o, to automatically parse the sentence with the following prompt:
>         ```
>         Extract and list all noun phrases that are likely found in the image that the following text describes, without any additional explanation or description:
>
>         The image depicts a cozy living room with a large flat-screen TV mounted on the wall. The room is furnished with a dining table surrounded by chairs, and a couch is placed nearby. A woman is standing in the room, possibly preparing to cook or serve a meal... [remaining model response]
>         ```
>         GPT-4o would response with the correct set of nouns that may be grounded in the image:
>         ```
>         - cozy living room
>         - large flat-screen TV
>         - wall
>         - dining table
>         - chairs
>         ... [remaining noun phrases of interest]
>         ```
>         Utilizing an LLM would lead to more accurate and advanced sentence parsing, but we chose the simple tool spaCy to avoid unnecessary complexity.
>         - We have revised Section 3.2 to better reflect that the usage of spaCy is not always necessary.
>
> - In addition, as the reviewer suggests, we have incorporated the discussion of the previous round into the paper:
>     - We have revised the abstract, Sections 1, 3.2, and 4.1 to consistently present our attend-and-segment based on point prompts and avoid confusions about attention accuracy.
>     - New results on RES and PNG are included in the main paper (see Section 4.3).
>     - Direct comparison with F-LMM is provided along with the results on RES and PNG (same as above, see Section 4.3).
>     - We have revised the presentation of the experiment section, to better reflect the analytical purpose of the instance segmentation experiment (Section 4.1 and the beginning of Section 4).
>     - We have provided visualized results to demonstrate that LMMs’ attention can correctly capture the semantics in complex scenarios (Figure 8 in Appendix D).
>
> Once again, we sincerely thank the reviewer for providing us with the valuable feedback for strengthening our work. We have updated the manuscript as suggested by the reviewer, and would like to request a re-evaluation of our work. We are more than glad to discuss if the reviewer has remaining concerns, but also would like to kindly remind the reviewer that the deadline for updating the manuscript is very soon, so it would be best if we could address further questions earlier.
>
> [Ref3] Deitke et al. Molmo and PixMo: Open Weights and Open Data for State-of-the-Art Multimodal Models. arXiv 2024.

---

> > ### Author Response · Authors · 2024-12-02
> > **Reminder About Reviewer Response Deadline**
> >
> > Dear Reviewer 4QCn,
> >
> > We sincerely appreciate your time and effort in reviewing our submission. Thank you once again for your valuable and constructive comments, and we have incorporated the results of our discussion into the revised manuscript.
> >
> > Given that the last day that reviewers can post responses is today (December 2, AoE), we would like to know if you have any remaining comments. We would be more than glad to have further discussion before the current phase ends.
> >
> > Best Regards,
> >
> > Authors of Submission 2465

---

> > > ### Author Response · Authors · 2024-12-03
> > > **Reminder About Reviewer Response Deadline**
> > >
> > > Dear Reviewer 4QCn,
> > >
> > > We sincerely thank you for your careful review of our work and the feedback that you have provided. We greatly value your comments, and we hope that our responses and updated manuscript have effectively addressed your remaining concerns.
> > >
> > > As the discussion period is nearing its end very soon, we would like to remind the you that if there are any remaining concerns requiring clarification, please do not hesitate to leave a comment. Should you find that our previous responses have satisfactorily addressed your concerns, we would be truly grateful if you could consider a higher rating, as it is crucial to the final evaluation of our submission.
> > >
> > > Thank you once again for your time and consideration.
> > >
> > > Best Regards,
> > >
> > > Authors of Submission 2465

---

### Official Review · Reviewer_r6Gw · 2024-10-31

**Soundness:** 3
**Presentation:** 3
**Contribution:** 3
**Rating:** 5
**Confidence:** 5

**Summary:**

This paper exploits geometric and spatial cues within the word-image attentions of large multimodal models (LMMs), which is conducive to visual grounding in multimodal conversations. Then the authors introduce the attend-and-segment paradigm for grounded conversation. Specifically, the word-image attention maps are normalised and fed to the SAM model for mask prediction. In addition, the authors propose to apply diffusion U-Net as visual encoders for multimodal models, which helps enhance visual grounding performance. The proposed method is evaluated at the Grounded Conversation Generation (GCG) benchmark.

**Strengths:**

1. The paper is well-organised and easy to follow. The method and experimental settings are all clearly illustrated.
2. The proposed method alleviates the need for pixel-level supervision for grounded conversation and preserves the excellent visual question-answering capability of modern LMMs.

**Weaknesses:**

1. The proposed attend-and-segment paradigm relies on the external tool SpaCy to parse noun phrases, which would simply enumerate all noun words and phrases from a sentence, regardless of whether the words correspond to visual objects in the images. Therefore, the parsing process can be quite noisy and might require prior human knowledge to filter out undesired nouns.

2. The proposed method is only evaluated on the GCG task, lacking experimental verification under tasks like referring segmentation which has been widely adopted in existing grounding LMMs.

3. The ad-hoc nature of using diffusion U-Net for enhanced grounding ability. Although the method circumvents additional grounding supervision, it requires re-training an LMM to incorporate the U-Net features on the million-scale pre-training and instruction-tuning datasets.

**Questions:**

1. Can the authors provide more details in obtaining object words / phrases by SpaCy? For example, how the nouns irrelevant to visual objects are filtered out?

2. Since the U-Net is already trained with cross-attention modules that interact with texts, would the word-image attention maps inside the U-Net be more capable of visual grounding than the attentions in the LLM part? If so, should we do visual grounding using u-net only instead of extracting LLM's attention maps?

---

> ### Author Response · Authors · 2024-11-25
> **Response to Reviewer r6Gw (Part 1)**
>
> We appreciate the detailed feedback you provided for our submission. We are encouraged by your acknowledgement of our “well-organised and easy to follow” writing, “all clearly illustrated” method and experimental settings, and alleviating “the need for pixel-level supervision for grounded conversation.” We provide the following clarifications in response to your concerns:
>
> *W1. The proposed attend-and-segment paradigm relies on the external tool SpaCy to parse noun phrases, which would simply enumerate all noun words and phrases from a sentence, regardless of whether the words correspond to visual objects in the images. Therefore, the parsing process can be quite noisy and might require prior human knowledge to filter out undesired nouns.*
>
> In this work, we used a simple tool spaCy to parse model responses and demonstrate that they could already achieve strong grounding capabilities. We would like to underscore this significant finding of implicitly learned grounding, **even using a simple NLP tool without complicated filtering mechanisms**.
>
> - For the GCG task (Table 2), we do not filter out any noun phrases for simplicity.  As shown by the higher performance on the grounded conversation generation (GCG) task in Table 2, this simple design already achieved strong grounding results.
>
> - In the instance segmentation analysis (Table 1), since we already know the 80 target object categories in MS-COCO, we 1) compute the embeddings of the category labels and the noun phrases with spaCy, 2) compute the cosine similarities between label embeddings and noun phrase embeddings, and 3) remove noun phrases whose similarities with all labels are lower than 0.5. This automated procedure was introduced in Line 323-346, Section 4.1. In other tasks, we can also set such “seed nouns” that can be grounded and remove undesired nouns that are dissimilar to them.
>
> - Furthermore, our method is not tied with spaCy, and we could utilize more advanced tools like large language models (LLMs) to automatically filter undesired nouns. For instance, we may provide the model response (image from https://cocodataset.org/#explore?id=139) to GPT-4o with the following prompt:
>     ```
>     Extract and list all noun phrases that are likely found in the image that the following text describes, without any additional explanation or description:
>
>     The image depicts a cozy living room with a large flat-screen TV mounted on the wall. The room is furnished with a dining table surrounded by chairs, and a couch is placed nearby. A woman is standing in the room, possibly preparing to cook or serve a meal... [remaining model response]
>     ```
>
>     GPT-4o responds with the correct set of nouns that may be grounded in the image:
>     ```
>     - cozy living room
>     - large flat-screen TV
>     - wall
>     - dining table
>     - chairs
>     ... [remaining noun phrases of interest]
>     ```
>     which does not include nouns that cannot be grounded (e.g., “The image”). Based on this automatic noun extraction and filtering, we can achieve accurate grounding and interactive conversation with users.
>
> *W2. The proposed method is only evaluated on the GCG task, lacking experimental verification under tasks like referring segmentation which has been widely adopted in existing grounding LMMs.*
>
> We greatly appreciate your suggestions on additional evaluation on other tasks. We have performed experiments on referring expression segmentation (RES) and panoptic narrative grounding (PNG) and achieved stronger performance compared with prior zero-shot methods that are also not trained with explicit grounding supervision. For example, our approach achieves 45.3 average recall in PNG, outperforming the previous zero-shot state-of-the-art method by 6.8 average recall. The details are included in Section D of the updated manuscript. The results on RES and PNG further demonstrate that LMMs can implicitly learn to accurately perform visual grounding without explicit grounding supervision, and our approach is the key to unlock such grounding ability.

---

> > ### Author Response · Authors · 2024-11-25
> > **Response to Reviewer r6Gw (Part 2)**
> >
> > *W3. The ad-hoc nature of using diffusion U-Net for enhanced grounding ability. Although the method circumvents additional grounding supervision, it requires re-training an LMM to incorporate the U-Net features on the million-scale pre-training and instruction-tuning datasets.*
> >
> > - We would like to emphasize that our usage of diffusion models **is not “ad-hoc,”** but rather well-rooted in the LMM literature. As demonstrated by various models (e.g., LISA and GLaMM), visual grounding is an important application of LMMs. The community has recently realized the limitations of the default CLIP visual encoder, especially in grounding, and started some initial exploration [Ref1-3]. Our experiments (Table 1, 2, and 5) find that diffusion models, strong at both vision-language alignment and grounding, can serve as a stronger visual encoder for improved grounding than CLIP and DINOv2, so we adopt them in our DiffLMM model.
> >
> > - In terms of training, DiffLMM does not require specialized training data and can be applied in general scenarios of LMMs. To demonstrate this, we adopted the same training data and procedure as LLaVA-1.5, and the training cost was almost the same as the original LLaVA-1.5 (about 13.5 hours in total on 8 NVIDIA A100 GPUs). This cost is **significantly smaller than** grounding LMMs (e.g., running the finetuning stage of GLaMM already requires 20 hours).
> >
> > - Also, many of our results are acquired by applying our attend-and-segment method on existing LMMs, **without any retraining at all** (see LLaVA-1.5, Cambrian-1, and LLaVA-NeXT in Table 1 and Table 2).
> >
> > *Q1. Can the authors provide more details in obtaining object words / phrases by SpaCy? For example, how the nouns irrelevant to visual objects are filtered out?*
> >
> > Please check our response for Weakness 1. We use spaCy to parse sentences into noun phrases (https://spacy.io/usage/linguistic-features#noun-chunks). In our experiments on GCG, we do not filter any noun phrases. For instance segmentation, since we already know the target object labels, we can compute embedding similarities between noun phrases and the target labels, and filter out noun phrases with low similarities.
> >
> > *Q2. Since the U-Net is already trained with cross-attention modules that interact with texts, would the word-image attention maps inside the U-Net be more capable of visual grounding than the attentions in the LLM part? If so, should we do visual grounding using u-net only instead of extracting LLM's attention maps?*
> >
> > - We observe that the word-image attention maps inside the U-Net are more noisy than the LLM attention maps, and thus cannot be effectively utilized for visual grounding. We visualize the attention maps inside the U-Net in Figure 9 of the updated manuscript, which can be compared with Figure 8 that visualizes the LLM attention maps. From the visualization of the U-Net attention maps, we observe less clear patterns and weaker correspondence between words and objects in the image. Therefore, they are not suitable for visual grounding.
> >
> > - Furthermore, attention maps from the U-Net suffer from two additional limitations:
> >
> >     - The text has to be given to the model as conditioning. In contrast, LMMs can generate text on their own, and use its own text for visual grounding.
> >
> >     - The text encoder (from CLIP) has a limited capacity and context length (fixed to 77 tokens). Therefore, the text-image correspondence is weak and cannot be extended to long text responses.
> >
> > [Ref1] Tong et al. Cambrian-1: A Fully Open, Vision-Centric Exploration of Multimodal LLMs. arXiv 2024.
> >
> > [Ref2] Zong et al. MoVA: Adapting Mixture of Vision Experts to Multimodal Context. NeurIPS 2024.
> >
> > [Ref3] Ranzinger et al. AM-RADIO: Agglomerative Vision Foundation Model Reduce All Domains Into One. CVPR 2024.

---

> ### Comment · Reviewer_r6Gw · 2024-11-26
> **Reviewer Response (1)**
>
> Thanks for the authors' diligent efforts to resolve reviewers' concerns. For now, I am quite split on the final assessment.
>
> On the one hand, my concerns about the parsing approaches have been well addressed, showing quite reasonable parsing results from LLMs like GPT4-O. The authors also provided quantitative results on more grounding benchmarks, including PNG and RES.
>
> On the other hand, I would like to insist on my concerns about using the diffusion unets, or employing any other visual backbone to improve visual grounding. Such design choices indicate that the grounding ability of existing LMMs is too weak to generate reasonable results, which requires additional re-training of an LMM with a third-party visual module. From my experience, the resources in re-training an LMM can be comparable to an LMM with grounding supervision. And the resources used in building modern LMMs are also increasing (as we have seen in Llava-Next/Llava-One-Vision and Cambrian).
>
> I am hanging over between 5 and 6. And I wish to see the feedback from other viewers and ACs.

---

> > ### Author Response · Authors · 2024-11-26
> > **Response to Reviewer r6Gw Follow-up**
> >
> > Thank you very much for reading our responses and acknowledging that we have addressed the previous concerns on parsing and more visual grounding tasks.
> >
> > Regarding your remaining concern on the adoption of diffusion models, we would like to highlight that we proposed DiffLMM as one solution for stronger grounding capabilities without additional training data. The other major contribution of our work is attend-and-segment, an approach to unlocking the grounding ability of **any existing LMMs without retraining**. Retraining LMMs is not mandatory in our work.
> >
> > - We did not mean to imply that “the grounding ability of existing LMMs is too weak to generate reasonable results.” In fact, **supported by our attend-and-segment method, the original LLaVA-1.5 already achieves a higher mask recall in the GCG task than the extensively supervised LMMs** (Table 2, 44.2 recall of LLaVA-1.5 + a&s vs. 41.8 recall of GLaMM).
> >
> > - There are other efforts in enhancing LMMs in the community, and our approach is compatible with them. We also **showed results of applying our attend-and-segment method on more recent LMMs without any retraining**. In Table 2, LLaVA-NeXT and Cambrian-1 are both strong in visual grounding, with the support of attend-and-segment. More specifically:
> >     - Both LLaVA-NeXT and Cambrian-1 achieve higher mIoU than LLaVA-1.5, indicating better segmentation quality for each visual entity.
> >     - LLaVA-NeXT has a higher mask recall than LLaVA-1.5, which is also on par with DiffLMM.
> >     - Cambrian-1 has a slightly lower mask recall, but still comparable with the grounding-supervised LMM GLaMM. We believe that the slightly lower mask recall is due to the fact that Cambrian-1 tends to generate shorter responses with the default setting (see the discussion in Sec 5.3 of the Cambrian-1 paper).
> >
> > - DiffLMM achieves the overall best grounding results in our work with minimal changes to the original LLaVA-1.5. For improved vision-language comprehension, there needs to be some modifications of the standard backbone, and then a retraining is usually inevitable. For instance, LLaVA-NeXT adopts dynamic resolution, and Cambrian-1 uses an ensemble of four visual backbones. Both of them use larger datasets for training than LLaVA-1.5. In our case of DiffLMM, we merely enhance the visual encoding with diffusion models for better grounding, and no other changes are made to the training data or procedure. Therefore, we believe our modification to LLaVA-1.5 is minimal.
> >
> > To conclude, we propose attend-and-segment for grounding existing LMMs that are trained without explicit grounding supervision, and DiffLMM for enhanced implicit grounding with minimal changes to LLaVA-1.5. Attend-and-segment does require any retraining and can be directly applied on recent LMMs (e.g., LLaVA-NeXT and Cambrian-1) to achieve strong visual grounding.

---

> > > ### Author Response · Authors · 2024-11-28
> > > **Response to Reviewer r6Gw**
> > >
> > > Thank you for the previous feedback for improving our work. We have provided another response and updated the manuscript, trying to address the remaining concern of the reviewer. We would like to kindly remind the reviewer that today is the last day that we can make changes to the PDF document, so it would be greatly appreciated if the reviewer could confirm whether the remaining concern has been addressed. Thank you again for the time and effort in providing helpful feedback for our work.

---

> > > > ### Author Response · Authors · 2024-12-02
> > > > **Reminder About Reviewer Response Deadline**
> > > >
> > > > Dear Reviewer r6Gw,
> > > >
> > > > We sincerely appreciate your time and effort in reviewing our submission. Thank you once again for your valuable and constructive comments, and we have incorporated the results of our discussion into the revised manuscript.
> > > >
> > > > Given that the last day that reviewers can post responses is today (December 2, AoE), we would like to know if you have any remaining comments. We would be more than glad to have further discussion before the current phase ends.
> > > >
> > > > Best Regards,
> > > >
> > > > Authors of Submission 2465

---

> > > > > ### Author Response · Authors · 2024-12-03
> > > > > **Reminder About Reviewer Response Deadline**
> > > > >
> > > > > Dear Reviewer r6Gw,
> > > > >
> > > > > We sincerely thank you for your careful review of our work and the feedback that you have provided. We greatly value your comments, and we hope that our responses and updated manuscript have effectively addressed your remaining concerns.
> > > > >
> > > > > As the discussion period is nearing its end very soon, we would like to remind the you that if there are any remaining concerns requiring clarification, please do not hesitate to leave a comment. Should you find that our previous responses have satisfactorily addressed your concerns, we would be truly grateful if you could consider a higher rating, as it is crucial to the final evaluation of our submission.
> > > > >
> > > > > Thank you once again for your time and consideration.
> > > > >
> > > > > Best Regards,
> > > > >
> > > > > Authors of Submission 2465

---

### Official Review · Reviewer_hPdE · 2024-11-04

**Soundness:** 3
**Presentation:** 2
**Contribution:** 3
**Rating:** 5
**Confidence:** 4

**Summary:**

This paper aims to exploit the grounding ability of large multimodal models (LMMs). Unlike the traditional methods that finetunes LMMs using grounding supervision, this work introduces attend-and-segment to leverage attention maps from LMMs for pixel-level segmentation. Diffusion-based visual encoder is applied to capture both high-level and low-level visual features. The experimental results demonstrate the proposed DiffLMM achieves remarkable performance across different tasks, especially on the grounded conversation generation (GCG) task.

**Strengths:**

This work exploits the grounding ability of LMMs without explicit grounding supervision. Motivated by the attention maps from LMMs capturing the region of interest, attend-and-segment mechanism is proposed to find out the segmentation masks. The proposed DiffLMM achieves the impressive results on the grounded conversation generation task.

**Weaknesses:**

As the core motivation of this work, the efficacy of attention maps from LMMs should be validated. 1) The precision and recall of attention maps representing objects are expected to be reported, which indicate whether they find out the regions of interest and whether the irrelevant objects/background are detected respectively. 2) Employing a single point with the highest attention value may miss to hit multiple objects or multiple regions of interest. It is unclear how to deal with these situations and how good this design is for capturing the most relevant object or region. 3) The attention maps from different LMMs and different prompts are also worth exploration.

Applying the diffusion model as the visual encoder is claimed as another contribution of this paper. However, as shown in Table 5, the performance of DiffLMM using SD-1.5 + CLIP is even inferior to that of original LLaVA-1.5 using CLIP backbone without PE and IC, raising concerns regarding the effectiveness and necessity of diffusion-based visual encoder.

The applied SAM prompt (mask or point) is a critical technical implementation. This technical design should be specified in the method section, rather than only mentioned in the ablation study. Also, it is expected to be clarified if SAM prompts differ for different tasks.

The verification of scalability of DiffLMM is absent. To evaluate the scalability of the proposed method, experiments involving data scaling up are expected to be provided.

To extensively demonstrate both the grounding and conversation/reasoning abilities of DiffLMM, it would be better to also include the experimental results on tasks such as referring expression segmentation, panoptic narrative grounding and reasoning segmentation.

**Questions:**

Please see weaknesses.

---

> ### Author Response · Authors · 2024-11-25
> **Response to Reviewer hPdE (Part 1)**
>
> We appreciate the detailed feedback you provided for our submission. We are encouraged by your acknowledgement of our discovery of the “grounding ability of LMMs without explicit grounding supervision,” our attend-and-segment mechanism being able to “find out the segmentation masks,” and our “impressive results on the grounded conversation generation task.” We provide the following clarifications in response to your concerns:
>
> *W1.1. As the core motivation of this work, the efficacy of attention maps from LMMs should be validated. The precision and recall of attention maps representing objects are expected to be reported, which indicate whether they find out the regions of interest and whether the irrelevant objects/background are detected respectively.*
>
> We would like to clarify that our attend-and-segment method **only uses the point with the highest attention value** from the attention map for prompting SAM. In fact, we do not require the entire attention map to be precise, as long as it is able to produce a reasonable starting point for SAM as a prompt. Therefore, we focus on validating the efficacy of the **prompt point produced by the attention map** in our experiments. In our pilot study of instance segmentation (Table 1), we already validated that our prompt extracted from LMM attention maps could provide effective grounding, **even without SAM**:
>
> - The PAcc metric computes the ratio of prompt points derived from attention maps that fall into the masks of the correct object category, which exactly reflect **“whether the irrelevant objects/background are detected.”**
>
> - The mask AR metric reflects the ratio of **regions of interest that can be discovered by the attention maps**. Note that even if the prompt point falls into the correct region, SAM may not provide a perfect mask, so this AR metric is a lower bound of the actual object recall. Even so, we reach over 45% recall on large objects. In other words, more than 45% large objects can be discovered by the attention maps. We notice that the recall on small objects is low, which is due to the relatively low resolution (336 x 336) in the current LLaVA-based paradigm.
>
> *W1.2. Employing a single point with the highest attention value may miss to hit multiple objects or multiple regions of interest. It is unclear how to deal with these situations and how good this design is for capturing the most relevant object or region.*
>
> Although a single point usually only locates one object at a time, we observe that our approach is still **able to locate multiple objects of the same category in a complex scene**.
>
> - Different from grounding LMMs that typically generate one mask per query, our approach is able to generate one grounding mask for every noun phrase in the response. In other words, our approach is not limited to capturing just one object or region in each response.
>
> - Therefore, when generating a description of a complex scene, the LMM can describe multiple similar objects separately, and then our attend-and-segment method produces segmentation masks for all individual objects that are mentioned in the model response. For example, in Figure 8 of the updated manuscript, we visualize examples where the attention map can correctly differentiate one specific object from other similar objects and locate it.
>
> *W1.3. The attention maps from different LMMs and different prompts are also worth exploration.*
>
> In Table 1, we already compared various LMMs including LLaVA (with CLIP/ConvNeXt CLIP/DINOv2 backbones), our DiffLMM, and Cambrian-1. Different prompts affect the output token sequence, and thus change the final grounding results. Per the reviewer’s request, we try three different prompts in the instance segmentation analysis with DiffLMM with SD-1.5+CLIP backbone (see Table 1 in the main paper):
> - Prompt 1 (Standard, the same as in the original experiments): “Describe the image in detail.”
> - Prompt 2 (Concise): “Briefly describe the image in a few words.”
> - Prompt 3 (Focusing on large objects): “Focus on the large objects in this image and describe them.”
>
> The results are summarized as below:
>
> | Prompt ID | PAcc | AP$\_S$ | AP$\_M$ | AP$\_L$ | AP | AR$\_S$ | AR$\_M$ | AR$\_L$ | AR |
> | :---- | :---- | :---- | :---- | :---- | :---- | :---- | :---- | :---- | :---- |
> | 1 | 40.22 | 1.6 | 7.9 | 9.6 | 5.6 | 6.3 | 25.5 | 47.3 | 26.0 |
> | 2 | 49.93 | 1.7 | 7.4 | 14.4 | 7.5 | 3.3 | 15.3 | 36.8 | 18.1 |
> | 3 | 41.77 | 1.3 | 7.3 | 15.5 | 5.6 | 4.4 | 23.4 | 49.4 | 24.6 |
>
> In general, prompts can change the behavior of the model response, and thus influence how attention maps can be leveraged for object grounding. The second prompt leads to shorter responses and higher precision but lower recall. The third prompt focuses on large objects and is more beneficial for locating such objects.

---

> > ### Author Response · Authors · 2024-11-25
> > **Response to Reviewer hPdE (Part 2)**
> >
> > *W2. Applying the diffusion model as the visual encoder is claimed as another contribution of this paper. However, as shown in Table 5, the performance of DiffLMM using SD-1.5 + CLIP is even inferior to that of original LLaVA-1.5 using CLIP backbone without PE and IC, raising concerns regarding the effectiveness and necessity of diffusion-based visual encoder.*
> >
> > We introduced the diffusion-based visual encoder mainly for **enhancing** the implicit grounding capability, as already demonstrated by the improvements of +6.21 PAcc in Table 1 (instance segmentation) and +4.4 Recall in Table 2 (grounded conversation generation). Meanwhile, we needed DiffLMM to **preserve** general vision-language conversation abilities, as shown in Table 3.
> >
> > - Our work mainly focuses on overcoming the limitations of prior grounding LMMs, which could not solve VQA tasks (Line 420-428, Section 4.3) due to catastrophic forgetting during finetuning for visual grounding (Line 077-070, Section 1). Improving LLaVA on general VQA benchmarks is not our objective.
> >
> > - Table 5 is comparing the pretraining loss, as a proxy for vision-language alignment and VQA tasks. The gap between LLaVA and DiffLMM is minor, consistent with the comparison in Table 3. We would like to point out that introducing more visual encoders does not always lead to **better performance in all tasks** due to discrepancies between features. The results are consistent with previous observations in Cambrian-1 [Ref1] (see Table 3) and MoVA [Ref2] (see Table 1).
> >
> > To conclude, supported by the necessary diffusion-based visual encoder, DiffLMM has already effectively achieved its goal of enhanced grounding and preserved conversation capabilities, overcoming the limitations of prior grounding LMMs that cannot perform VQA tasks (e.g., LISA and GLaMM).
> >
> > *W3. The applied SAM prompt (mask or point) is a critical technical implementation. This technical design should be specified in the method section, rather than only mentioned in the ablation study. Also, it is expected to be clarified if SAM prompts differ for different tasks.*
> >
> > - In Section 3.2 Line 236 of the original manuscript, we already mentioned that “For each token that requires grounding, we produce its corresponding binary mask by prompting SAM with the coordinate that has the highest normalized attention.” In other words, we find the point with the highest attention value, and use its coordinate as a point prompt to SAM.
> >
> > - We consistently used point prompts in all experiments, except the ablation study that compared point vs. mask prompts. We have revised the manuscript to clarify this SAM prompting strategy.
> >
> > *W4. The verification of scalability of DiffLMM is absent. To evaluate the scalability of the proposed method, experiments involving data scaling up are expected to be provided.*
> >
> > We understand that the reviewer is interested in how DiffLMM scales with data, but would like to seek further clarification from the reviewer about the **necessity of such experiments**.
> >
> > - Our DiffLMM adopts exactly the same training data and training recipe as the well-established LLaVA-1.5 model, which can be considered as a standard practice in the LMM literature.
> >
> > - Furthermore, we do not observe such data scaling experiments previously conducted in grounding LMMs (e.g., LISA, GLaMM, Kosmos-2).
> >
> > To our best understanding of the reviewer’s comments, we retrain our DiffLMM models on smaller subsamples of the instruction tuning data of LLaVA-1.5, evaluate them on the VQAv2 benchmark, and compare them with the original LLaVA-1.5. We observe similar performance trends in the two models.
> >
> > | Model | 25% Data | 50% Data | 100% Data |
> > | :---- | :---- | :---- | :---- |
> > | LLaVA-1.5 | 74.0 | 75.4 | 78.5 |
> > | DiffLMM | 73.9 | 75.7 | 78.3 |
> >
> > We would be glad to have further discussion if the reviewer could provide some clarifications on the scalability verification.

---

> > > ### Author Response · Authors · 2024-11-25
> > > **Response to Reviewer hPdE (Part 3)**
> > >
> > > *W5. To extensively demonstrate both the grounding and conversation/reasoning abilities of DiffLMM, it would be better to also include the experimental results on tasks such as referring expression segmentation, panoptic narrative grounding and reasoning segmentation.*
> > >
> > > We greatly appreciate your suggestions on additional evaluation on other tasks. We have performed experiments on referring expression segmentation (RES) and panoptic narrative grounding (PNG) and achieved stronger performance compared with prior zero-shot methods that are also not trained with explicit grounding supervision. For example, our approach achieves 45.3 average recall in PNG, outperforming the previous zero-shot state-of-the-art method by 6.8 average recall. The details are included in Section D of the updated manuscript. The results on RES and PNG further demonstrate that LMMs can implicitly learn to accurately perform visual grounding without explicit grounding supervision, and our approach is the key to unlock such grounding ability.
> > >
> > > [Ref1] Tong et al. Cambrian-1: A Fully Open, Vision-Centric Exploration of Multimodal LLMs. arXiv 2024.
> > >
> > > [Ref2] Zong et al. MoVA: Adapting Mixture of Vision Experts to Multimodal Context. NeurIPS 2024.

---

> > > > ### Comment · Reviewer_hPdE · 2024-11-26
> > > >
> > > > Thanks for the detailed feedback from the authors, which clarifies some confusions. However, I believe the paper requires a major revision and may not be ready for publication at this stage. For example:
> > > >
> > > > The motivation of this work is that the attention maps from LMMs naturally imply the grounding information as presented in the abstract and introduction sections, while employing a single point with the highest value of the attention maps is only a specific operation to filter out some false positives. Therefore, the efficacy of attention maps is critical to the standing point of this paper. Otherwise, the sections of abstract and introduction are expected to be modified, in order to avoid the readers' interest in the LLMs' original attention maps.
> > > >
> > > > The ablation study in Table 5 compares the performance of DiffLMM using SD-1.5 + CLIP and LLaVA-1.5 using CLIP as visual backbone. However, the rebuttal mentions the task in Table 5 is not the main objective of this work. If so, I believe it requires a new setting for the ablation study.
> > > >
> > > > For the necessity of scalability, it is because the paper aims to explore the grounding ability of LMMs without explicit grounding supervision. Apart from preserving the conversation ability, the proposed method reduces the costs of object-level annotations. Also, scalability is the primary problem mentioned in the third paragraph of the introduction section. So the experiments to showcase the scalability would highlight the advantage of the proposed method.
> > > >
> > > > As all the reviewers mentioned, the results on widespread benchmarks are vital to demonstrate the grounding and conversation abilities of DiffLMM. The suggested experiments on tasks such as referring expression segmentation, panoptic narrative grounding and reasoning segmentation should be included in the main text.
> > > >
> > > > Nonetheless, I do believe the significance of exploiting the grounding ability of LMMs without explicit grounding supervision, and strongly recommend the continued research in this direction.

---

> > > > > ### Author Response · Authors · 2024-11-27
> > > > > **Response to Reviewer hPdE Follow-up**
> > > > >
> > > > > We greatly value the response from the reviewer, and we are encouraged that we have clarified most of the reviewer’s previous confusions. We also sincerely appreciate the reviewer acknowledging “the significance of exploiting the grounding ability of LMMs without explicit grounding supervision,” which is indeed the core contribution of our work.
> > > > >
> > > > > We welcome the suggestions on the paper revision, which would significantly enhance our work. It seems that the reviewer’s remaining concerns are about integrating the clarifications and newly added results into the revision. Therefore, we have revised the manuscript to incorporate the results from the discussion, to clear the remaining concerns from the reviewer.
> > > > >
> > > > > - We have revised the abstract, introduction, and method sections to introduce our attend-and-segment method with a more consistent presentation, focusing on the **point-based prompts derived from the attention within LMMs**.
> > > > >
> > > > > - Regarding the ablation study that compares DiffLMM with LLaVA-1.5 (Table 5 in the original manuscript,  Table 7 in the updated version), we would like to clarify:
> > > > >     - **Preserving (not improving)** general conversation capabilities is also a major objective of our work in parallel to improving grounding, so we implemented this ablation study to understand the best configuration for DiffLMM. In our last response, we mentioned that “improving LLaVA on general VQA benchmarks is not our objective.” However, we would like to clarify that this should not be interpreted as “preserving general conversation capabilities is not the main objective.” We indeed aimed to achieve comparable VQA performance as LLaVA-1.5, but did not expect to outperform it.
> > > > >     - Since we already extensively compared the grounding capabilities of DiffLMM and LLaVA-1.5 in visual grounding experiments (instance segmentation, GCG, RES, and PNG, see Tables 1-4), we focus on the comparison of vision-language alignment in this ablation study.
> > > > >
> > > > >     We have revised the introduction of the ablation study (Section 4.5 and Table 7) on DiffLMM to better reflect its objective on general conversations. We hope this clarifies the reviewer’s concern.
> > > > >
> > > > > - We have revised the introduction, adjusting the explanation of the first constraint of grounding LMMs **from “limited scalability” to “limited training data,”** which more accurately aligns with the objective of our work—removing the need for grounding supervision.
> > > > >
> > > > >     In response to the reviewer’s remaining concern about scalability, we would like to clarify:
> > > > >     - Our attend-and-segment method can be seamlessly integrated with existing LMMs (e.g., LLaVA-1.5, LLaVA-NeXT, Cambrian-1) without retraining, which directly benefits from the improved scales of recent LMMs.
> > > > >     - As for our DiffLMM, as explained in the previous response, it adopts exactly the same training data and training recipe as the well-established LLaVA-1.5 model, which can be considered as a standard practice in the LMM literature.
> > > > >     - Furthermore, we do not observe such data scaling experiments previously conducted in grounding LMMs (e.g., LISA, GLaMM, Kosmos-2).
> > > > >
> > > > > - We have incorporated the summarized results on RES and PNG in the main paper (Section 4.3 and Tables 3 and 4).
> > > > >
> > > > > Once again, we sincerely thank the reviewer for providing us with the valuable feedback for strengthening our work. We have updated the manuscript as suggested by the reviewer, and would like to request a re-evaluation of our work.

---

> > > > > > ### Author Response · Authors · 2024-11-28
> > > > > > **Response to Reviewer hPdE**
> > > > > >
> > > > > > Thank you for the previous comments and suggestions on revising our manuscript. We have updated the PDF document according to all reviewers’ feedback. We would like to kindly remind the reviewer that today is the last day that we can make changes to the PDF document, so it would be greatly appreciated if the reviewer could confirm whether our updated version addresses the remaining concerns. Thank you again for the time and effort in providing helpful feedback for our work.

---

> > > > > > > ### Author Response · Authors · 2024-12-02
> > > > > > > **Reminder About Reviewer Response Deadline**
> > > > > > >
> > > > > > > Dear Reviewer hPdE,
> > > > > > >
> > > > > > > We sincerely appreciate your time and effort in reviewing our submission. Thank you once again for your valuable and constructive comments, and we have incorporated the results of our discussion into the revised manuscript.
> > > > > > >
> > > > > > > Given that the last day that reviewers can post responses is today (December 2, AoE), we would like to know if you have any remaining comments. We would be more than glad to have further discussion before the current phase ends.
> > > > > > >
> > > > > > > Best Regards,
> > > > > > >
> > > > > > > Authors of Submission 2465

---

> > > > > > > > ### Author Response · Authors · 2024-12-03
> > > > > > > > **Reminder About Reviewer Response Deadline**
> > > > > > > >
> > > > > > > > Dear Reviewer hPdE,
> > > > > > > >
> > > > > > > > We sincerely thank you for your careful review of our work and the feedback that you have provided. We greatly value your comments, and we hope that our responses and updated manuscript have effectively addressed your remaining concerns.
> > > > > > > >
> > > > > > > > As the discussion period is nearing its end very soon, we would like to remind the you that if there are any remaining concerns requiring clarification, please do not hesitate to leave a comment. Should you find that our previous responses have satisfactorily addressed your concerns, we would be truly grateful if you could consider a higher rating, as it is crucial to the final evaluation of our submission.
> > > > > > > >
> > > > > > > > Thank you once again for your time and consideration.
> > > > > > > >
> > > > > > > > Best Regards,
> > > > > > > >
> > > > > > > > Authors of Submission 2465

---

### Author Response · Authors · 2024-11-25
**General Response to All Reviewers**

We appreciate the time and effort invested by all the reviewers in evaluating our manuscript and providing constructive suggestions. To provide better clarification, we have updated the manuscript by including additional experiment results on referring expression segmentation (RES) and panoptic narrative grounding (PNG) tasks and more visualized results. The updates are marked in blue.

We encourage the reviewers to check the updated manuscript while reading our direct responses to each reviewer. When comparing our approach with prior methods, we would like to highlight that our approach **does not require any explicit grounding supervision**, while previous grounding LMMs all require extensive grounding supervision to accomplish visual grounding tasks. Therefore, it would not be fair to directly compare the performance of ours with prior work in grounding LMMs.

We hope our responses can successfully clarify your previous concerns, and please let us know if further discussion is needed. Once again, we sincerely appreciate your valued feedback for improving our work.

---

### Author Response · Authors · 2024-11-27
**General Response to All Reviewers**

We thank all the reviewers for providing their constructive feedback on improving our work. We have updated the manuscript again to reflect the helpful comments made by all reviewers. The updated parts are marked in blue. We would like to invite all reviewers to check and re-evaluate our work. If further questions arise, we are more than glad to continue the discussion.

---

### Author Response · Authors · 2024-12-04
**Summary of Author-Reviewer Discussion (Part 1)**

We greatly value the constructive feedback provided by all the reviewers and we have endeavored to address all of their concerns. We summarize the author-reviewer discussion as follows:

**Strengths Acknowledged by Reviewers**

We appreciate the reviewers acknowledging 1) our motivating discovery of grounding ability of LMMs without explicit grounding supervision (Reviewers hPdE, r6Gw, 4QCn), 2) our impressive results on tasks like grounded conversation generation (Reviewers hPdE, 4QCn), 3) our preservation of the general VQA capabilities of modern LMMs (Reviewers r6Gw, 4QCn), and 4) our paper organization and presentation (Reviewers r6Gw, 4QCn).

All the reviewers recognize our core contribution of LMM grounding without explicit grounding supervision. In contrast to existing works on LMM grounding that rely on extensive grounding supervision and specialized grounding modules to perform visual grounding tasks (e.g., grounded conversation generation and panoptic narrative grounding), our work proposes the first solution that directly unlocks the grounding ability of modern LMMs without necessarily retraining them.

**Major Concerns and Discussions**

1. Reviewers suggested including results on additional visual grounding tasks such as referring expression segmentation (RES) and panoptic narrative grounding (PNG).
In the initial manuscript, we focused on the task of grounded conversation generation (GCG) for several reasons: 1) GCG is a more challenging and comprehensive task, testing both the generation and visual grounding abilities of LMMs, while RES and PNG only require visual grounding. 2) The state-of-the-art grounding LMM, GLaMM, proposed the GCG task and focused on its evaluation. We followed GLaMM and considered GCG as the main task for comparison. Nevertheless, following reviewers’ suggestions, we applied our proposed approach to RES and PNG and outperformed prior zero-shot methods in both tasks. The additional results demonstrate that **our approach is general, effective, and widely applicable in various visual grounding tasks**. We have revised the manuscript to include the new results.

    Reviewer r6Gw explicitly acknowledged that our results on RES and PNG addressed the previous concern. Reviewer hPDE suggested including the RES and PNG results in the main paper, and we have updated Section 4.3 as suggested. Reviewer 4QCn did not raise further concerns on these visual grounding tasks.

2. Reviewers hPdE and 4QCn had a concern about how to extract effective grounding information from LMMs’ attention maps, due to the noisiness of attention maps. In response to the concern, we clarified: 1) Our attend-and-segment strategy only requires one point with the highest attention value as a point prompt to a segmentation model, so we do not require the entire attention map to be precise. 2) In the manuscript, we already verified that the point prompts derived by attend-and-segment have reasonable precision before using SAM for segmentation (PAcc metric in Table 1). 3) SAM is semantic-free and cannot improve the visual grounding ability. Without a point prompt correctly locating the target object, SAM is unable to produce an accurate segmentation mask.

    Reviewers hPdE and 4QCn suggested a paper revision on the introduction of the usage of attention maps, and we have revised the Abstract, Section 1, Section 3.2, and Section 4.1 of our manuscript to present our attend-and-segment strategy more clearly and avoid confusion.

---

> ### Author Response · Authors · 2024-12-04
> **Summary of Author-Reviewer Discussion (Part 2)**
>
> 3. Reviewers hPdE and r6Gw questioned the necessity of integrating diffusion-based features in DiffLMM. We provided clarifications: 1) The main objective of DiffLMM is to improve the grounding ability and preserve the general conversation ability, rather than improving over LLaVA on VQA tasks. DiffLMM addresses the limitation of prevalent grounding LMMs (e.g., LISA and GLaMM), which lose the general conversation ability after grounding-specialized finetuning. 2) The grounding advantages have been shown in Tables 1-4, and the preservation of general VQA is demonstrated by Tables 5 and 7. 3) **Our main contribution, attend-and-segment, can be employed with existing LMMs (e.g., LLaVA-1.5, LLaVA-NeXT, and Cambrian-1) without retraining a diffusion-based model.**
>
>     Reviewer hPDe suggested a revision of the ablation study to better show the objective, and we have revised Section 4.5 and Table 7 as requested. Reviewer r6Gw replied about the concerns of using a different visual backbone for improving visual grounding and additional training costs. We have provided further explanation: 1) DiffLMM requires minimal changes to LLaVA-1.5 and uses the same training data, so the training cost is much lower than prevalent grounding LMMs (e.g., LISA and GLaMM). 2) Even without any retraining, our attend-and-segment strategy unlocks existing LMMs’ grounding ability, which already achieves better performance in GCG than GLaMM, as shown in Table 2.
>
> 4. Reviewers r6Gw and 4QCn had questions about the parsing process with the NLP tool spaCy, which simply enumerates all noun phrases without filtering or complex semantics. We clarified by explaining our straightforward usage of spaCy in the instance segmentation and GCG tasks, and provided examples of parsing model responses with more advanced tools like GPT-4o, and parsing complex phrases involving multiple nouns with spaCy.
>
>     Reviewer r6Gw expressed that our response resolved the concern. Reviewer 4QCn raised follow-up questions on attention map quality and direct comparison with prior methods that automatically generate targets. We have responded with additional discussions to explain that the attention map is indeed focusing on the correct targets, and we adopt the simplest tool to find noun phrases as targets automatically, which induces consistent comparison with prior methods.
>
> In summary, we are deeply grateful for the productive conversations with the reviewers and we believe that we have meticulously addressed all reviewer questions and **integrated the results of author-reviewer discussions into the manuscript revision**. **Our work, as the first work that discovers the implicit grounding ability in LMMs trained without grounding supervision**, is a valuable contribution to the LMM research community. We humbly request your favorable consideration of our submission. Again, we appreciate all the reviewers, ACs, SACs, and PCs for the time and devotion.

---

### Meta-Review · Area_Chair_an3j · 2024-12-21

**Metareview:**

This paper aims to exploit the grounding ability of large multimodal models (LMMs). Unlike the traditional methods that finetunes LMMs using grounding supervision, this work introduces attend-and-segment to leverage attention maps from LMMs for pixel-level segmentation. Diffusion-based visual encoder is applied to capture both high-level and low-level visual features. The experimental results demonstrate the proposed DiffLMM achieves remarkable performance across different tasks. The experiments are conducted on Grounded Conversation Generation (GCG) and VQA benchmarks.

Strengths:
+ The motivation of using attention maps for pixel-level grounding is reasonable.
+ The proposed DiffLMM achieves good experimental results compared to CLIP-based, and DINO-based methods for GCG, while maintaining general VQA performance.

Weaknesses:
+ The parsing step using SpaCy is noisy, which is not very convincing.
+ Some results of the proposed DIFF-LMM are unsatisfactory, limiting its applications in real scenarios.
+ More results on other benchmarks (such as referring expression segmentation, panoptic narrative grounding) are expected.

**Additional Comments On Reviewer Discussion:**

After the rebuttal, all reviewers still give negative ratings. There are some remaining concerns about this submission:
+ The writing and presentation should be further improved.
+ More reasonable ablation studies (such as different backbones) and more results on scalability are needed.
+ The usage of SpaCy.

In summary, I think there are still aspects can be improved for this submission, and recommend Reject.

---

### Decision · Program_Chairs · 2025-01-22

Reject